# LHM++: An Efficient Large Human Reconstruction Model for Pose-free Images to 3D

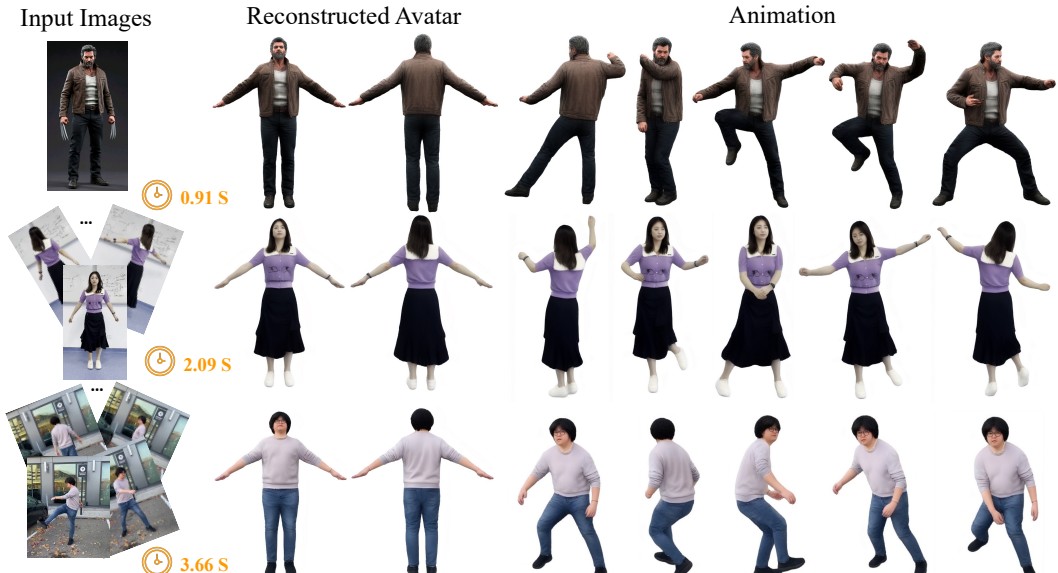

Figure 1. **3D Avatar Reconstruction and Animation Results of our LHM++.** Given a set of $N \geq 1$ images of a human subject, without requiring camera parameters or human pose annotations, our method can reconstruct a high-fidelity, animatable 3D human avatar in seconds.

## ABSTRACT

Reconstructing animatable 3D humans from casually captured images of articulated subjects without camera or pose information is highly practical but remains challenging due to view misalignment, occlusions, and the absence of structural priors. In this work, we present *LHM++*, an efficient large-scale human reconstruction model that generates high-quality, animatable 3D avatars within seconds from one or multiple pose-free images. At its core is an *Encoder–Decoder Point–Image Transformer* architecture that progressively encodes and decodes 3D geometric point features to improve efficiency, while fusing hierarchical 3D point features with image features through multimodal attention. The fused features are decoded into 3D Gaussian splats to recover detailed geometry and appearance. To further enhance visual fidelity, we introduce a lightweight 3D-aware neural animation renderer that refines the rendering quality of reconstructed avatars in real time. Extensive experiments show that our method produces high-fidelity, animatable 3D humans without requiring camera or pose annotations. Our code and models will be released to the public.

## 1 INTRODUCTION

Reconstructing high-quality, animatable 3D human avatars from casually captured images is a crucial task in computer graphics, with broad applications like virtual reality and telepresence. A practical solution should support rapid and robust reconstruction from minimal input—ideally using only

one or a few casually captured images, without relying on camera parameters, human pose annotations, or controlled capture environments. Such capability is essential for enabling scalable and accessible avatar generation in real-world scenarios.

Existing approaches to animatable 3D human reconstruction from monocular or multi-view videos typically rely on optimization-based frameworks that minimize photometric or silhouette reprojection losses Alldieck et al. (2018b); Weng et al. (2022); Guo et al. (2025). These methods usually require dozens or hundreds of images with accurate human pose estimation as a prerequisite. Moreover, the optimization process is often computationally expensive, taking several minutes or even hours to converge, thus limiting real-time applications.

More recently, LHM Qiu et al. (2025a), a feed-forward network for single-image 3D human reconstruction, has shown promising progress toward real-time performance. It employs a transformer-based architecture to fuse geometric point features initialized from the canonical SMPL-X surfaces and image features to directly predict a 3D Gaussian Splatting Kerbl et al. (2023) based avatar from a single image. However, as single-image-based methods are inherently limited by partial observations, they often struggle to reconstruct occluded or unseen regions, leading to oversmoothed surfaces or noticeable artifacts Saito et al. (2020); Zhuang et al. (2025).

A straightforward extension of LHM Qiu et al. (2025a) to multi-image settings would involve concatenating image tokens from multiple images and performing attention fusion. However, such a naive approach suffers from substantial memory and computational overhead due to the large number of geometric point features and the quadratic complexity of dense self-attention mechanisms.

In this work, we propose *LHM++*, a novel feed-forward framework for fast and high-fidelity 3D human reconstruction from one or a few images without requiring the camera and human poses. To achieve this, we design an efficient *Encoder-Decoder Point-Image Transformer (PIT) Framework* that hierarchically fuses 3D geometric features with multi-image cues. The framework is built upon *Point-Image Transformer blocks (PIT-blocks)*, which enable interaction between geometric and image tokens via attention fusion while maintaining scalability through spatial hierarchy.

We start by representing the SMPL-X anchor points as geometric tokens and extracting image tokens from each input image. The encoder stage comprises several PIT-blocks to progressively downsample the geometric tokens via Grid Pooling Wu et al. (2022). At each layer, the downsampled point tokens interact with image tokens through multimodal attention Esser et al. (2024), allowing compact yet expressive geometric representations to be enriched with visual information from multiple images. The decoder stage upsamples the geometric tokens to recover spatial resolution.

The resulting 3D geometry tokens are decoded to predict Gaussian splatting parameters, enabling high-quality rendering and animation. To further improve visual fidelity, we introduce a lightweight 3D-aware neural animation renderer, built on a DPT-head architecture, that refines the rendering quality of reconstructed avatars in real time. For robustness and generalization, we train our model on large-scale real-world human video datasets spanning diverse clothing styles, body shapes, and viewing conditions. In summary, our contributions are:

- We introduce *LHM++*, an efficient feed-forward model, capable of reconstructing high-quality and animatable 3D human avatars in seconds from one or a few casually captured images, without requiring either camera poses or human pose annotations.

- We propose a novel *Encoder-Decoder Point-Image Transformer* (PIT) architecture that hierarchically fuses 3D geometric point features and 2D image features using multimodal attention, enabling efficient and scalable integration of multi-image cues.

- Extensive experiments on both synthetic and real-world data demonstrate that LHM++ unifies single- and multi-image 3D human reconstruction, with superior generalization and visual quality.

## 2 RELATED WORK

**Human Reconstruction from A Single Image**  For single-image 3D human reconstruction, many methods adopt implicit neural representations Saito et al. (2019; 2020); Xiu et al. (2023); Cao et al. (2022); Zheng et al. (2021); Zhang et al. (2024b); Xiong et al. (2024); Yang et al. (2024) to model complex human geometries. To improve geometric consistency and generalizability, some

approaches Choutas et al. (2022); Kanazawa et al. (2018); Alldieck et al. (2018a; 2019) rely on parametric body models such as SMPL Loper et al. (2015); Pavlakos et al. (2019) to predict geometric offsets for the reconstruction of clothed humans. However, reconstruction from a single image is an ill-posed problem. Current cascade-type approaches Li et al. (2025); Wang et al. (2025d); Weng et al. (2024); Wang et al. (2025a); Qiu et al. (2025b) attempt to mitigate this issue by decoupling the process into two stages: multi-view image synthesis using generative models, followed by 3D reconstruction. While these methods require view-consistent generation in the first stage, which is often unstable and challenging, this ultimately affects the quality of the reconstruction.

Inspired by the success of large reconstruction models Hong et al. (2023); Tang et al. (2024a), emerging solutions aim to enable direct generalizable reconstruction through feed-forward networks which significantly accelerate the inference time. Human-LRM Weng et al. (2024) employs a feed-forward model to decode the triplane NeRF representation, then followed by a conditional diffusion-based novel views generation and reconstruction. IDOL Zhuang et al. (2025) introduces a UV-Alignment transformer model to decode Gaussian attribute maps in a structured 2D UV space. LHM Qiu et al. (2025a) leverages a Body-Head multimodal transformer architecture produces animatable 3D avatars with the face identity preservation and fine detail recovery. While these single-view methods often face challenges with occlusions and invisible regions, frequently resulting in geometrically implausible results or blurred textures.

Table 1. Comparison with state-of-the-art 3D human reconstruction methods. FF stands for Feed-forward, PF for Pose-free, and AM for Animatable.

| Method | # Image | FF | PF | AM | Runtime |
|---|---|---|---|---|---|
| CAR Liao et al. (2023) | 1 | ✗ | ✗ | ✔ | 5 Minutes |
| IDOL Zhuang et al. (2025) | 1 | ✔ | ✔ | ✔ | Seconds |
| AniGS Qiu et al. (2025b) | 1 | ✗ | ✔ | ✔ | 15 Minutes |
| LHM Qiu et al. (2025a) | 1 | ✔ | ✔ | ✔ | Seconds |
| Vid2Avatar Guo et al. (2023) | > 100 | ✗ | ✗ | ✔ | 1-2 Days |
| Hugs Kocabas et al. (2024) | > 80 | ✗ | ✗ | ✔ | 30 Minutes |
| Canonicalfusion Shin et al. (2024) | > 1 | ✗ | ✗ | ✔ | 11 Minutes |
| GPS-Gaussian Zheng et al. (2024) | 2 | ✔ | ✗ | ✗ | Seconds |
| 3DGS-Avatar Qian et al. (2024a) | > 20 | ✗ | ✗ | ✔ | 0.5 Hours |
| InstantAvatar Jiang et al. (2023) | > 20 | ✗ | ✗ | ✔ | Minutes |
| GaussianAvatar Hu et al. (2024) | > 20 | ✗ | ✗ | ✔ | 0.5-6 Hours |
| ExAvatar Moon et al. (2024) | > 20 | ✗ | ✗ | ✔ | 1.5-5 Hours |
| PuzzleAvatar Xiu et al. (2024) | 4 ∼ 6 | ✗ | ✔ | ✗ | 4-6 Hours |
| Vid2Avatar-Pro Guo et al. (2025) | > 100 | ✗ | ✗ | ✔ | Hours |
| Giga Zubekhin et al. (2025) | 1 ∼ 4 | ✔ | ✗ | ✔ | Seconds |
| FRESA Wang et al. (2025c) | 1 ∼ 4 | ✔ | ✔ | ✔ | Seconds |
| Ours | ≥ 1 | ✔ | ✔ | ✔ | Seconds |

sions and invisible regions, frequently resulting in geometrically implausible results or blurred textures.

Recently, HumanRAM Yu et al. (2025) adapts human reconstruction to novel view and pose synthesis using LVSM with static pose sparse view inputs, achieving photorealistic results but suffering from slow transformer-based rendering. The concurrent work, GIGA Zubekhin et al. (2025), employs UV map representations for generalizable reconstruction, but it relies on multi-view captures of the same action, complex camera setups, and motion calibration.

**Human Reconstruction from Monocular Videos** Video-based techniques further improve reconstruction consistency by using temporal cues. 4D replay methods Park et al. (2021); Weng et al. (2022) can reconstruct dynamic humans from monocular video or multiview video sequences, however, they are not able to drive the humans in novel poses since they do not build a standalone 3D model for humans. Therefore, a series of monocular video-based methods Jiang et al. (2022a); Qiu & Chen (2023); Hu & Liu (2024); Tan et al. (2025) build a static 3D human model and can drive the human in novel poses by binding the skinning weight.

Another series of works Jiang et al. (2023); Moon et al. (2024); Hu et al. (2024); Guo et al. (2025); Jiang et al. (2022b); Qian et al. (2024b); Yu et al. (2023); Zhan et al. (2025); Yang et al. (2025b); Kocabas et al. (2024) take it further by incorporating a 3D parametric human model into the optimization process, and thus can drive the human reconstruction in novel poses without any postprocessing. Despite impressive visual fidelity, they often require dozens of minutes and dozens of views for a good optimization, which limits their practical usage in real-world scenarios.

Unconstrained collection is ideal input for a practical application. However, existing methods Xiu et al. (2024); Yang et al. (2024) share a similar pipeline that uses a view generative model and score distillation sampling Poole et al. (2023) for shape optimization. As a result, they are costly for offline training and impractical for online reconstruction.

**Feed-Forward Scene Reconstruction** Recent years have witnessed a paradigm shift in geometric 3D vision, driven by the emergence of methods that eliminate traditional dependencies on camera calibration and multi-stage pipelines. At the forefront of this revolution lies the DUSt3R Wang et al. (2024) framework, which reimagines 3D reconstruction as a direct regression problem from image pairs to 3D pointmaps. By discarding the need for intrinsic camera parameters, extrinsic pose

Figure 2. **Overview of the proposed *LHM++*.** In the 2D space, we extract image tokens $\mathbf{T}_{\text{Img}}$ by DINOv2 from the input RGB images. In the 3D space, geometric tokens $\mathbf{T}_{\text{3D}}$ are represented by the MLP output of SMPL-X anchor points. Subsequently, we build our Encoder-Decoder Point-Image Transformer (PIT) to hierarchically fuse 3D tokens with 2D tokens, where the downsampled 3D tokens interact with 2D tokens via multimodal attention in each layer. The finalized 3D tokens are decoded to directly predict 3D Gaussian parameters, followed by a light-weight DPT-head for photorealistic animation.

estimation, or even known correspondence relationships, DUSt3R and its successors Yang et al. (2025a); Tang et al. (2024b); Lu et al. (2024); Wang et al. (2025b) have democratized 3D vision, enabling rapid reconstruction across diverse scenarios while achieving state-of-the-art performance in depth estimation, relative pose recovery, and scene understanding. However, general feed-forward reconstruction methods assume that images are captured from a static scene Wang et al. (2023); Li et al. (2024), while our LHM++ can accept human images with different camera and human poses as input and produce an animatable 3D avatar. The concurrent work, FastVGGT Shen et al. (2025), employs a token merge strategy to improve the efficiency of image token global attention.

## 3 METHOD

### 3.1 OVERVIEW

**Problem Formulation** Given a set of $N \geq 1$ RGB images $I^1, \ldots, I^N$ of a human subject, without known camera parameters or human pose annotations, our goal is to reconstruct a high-fidelity, animatable 3D human avatar $\chi$ in seconds.

We adopt the 3D Gaussians splatting (3DGS) Kerbl et al. (2023) as the representation, which allows for photorealistic, real-time rendering and efficient pose control. Each 3D Gaussian primitive is parameterized by its center location $\mathbf{p} \in \mathbb{R}^3$, directional scales $\boldsymbol{\sigma} \in \mathbb{R}^3$, and orientation (represented as a quaternion) $\mathbf{r} \in \mathbb{R}^4$. In addition, the primitive includes opacity $\rho \in [0, 1]$ and spherical harmonic (SH) coefficients $\mathbf{f}$ to model view-dependent appearance.

Inspired by LHM Qiu et al. (2025a), we employ a set of spatial points $P \in \mathbb{R}^{N_{\text{points}} \times 3}$ uniformly sampled from the SMPL-X surface in its canonical pose to serve as the anchors. Conditioned on the multi-image inputs, these points are processed and decoded to regress the human 3D Gaussian appearance in canonical space through a feed-forward transformer-based architecture. The pipeline can be formulated as:

$$\chi\{\mathbf{p}, \mathbf{r}, \mathbf{f}, \rho, \boldsymbol{\sigma}\} = \text{LHM++}(P \mid I^1, \ldots, I^N). \tag{1}$$

**Model Design** A straightforward solution to this problem is to extend LHM to support multiple image inputs by directly concatenating all available image tokens and performing attention operations between 3D point tokens and image tokens. However, this naive extension results in significant computational and memory overhead due to the quadratic complexity of self-attention operations with respect to the total number of tokens, i.e., $\mathcal{O}((N_{\text{points}} + N)^2)$.

To mitigate this issue, we explore strategies to reduce the number of geometric point tokens involved in attention. However, we empirically observe that simply reducing the number of point tokens significantly degrades reconstruction performance. To address this trade-off, we propose an efficient *Encoder-Decoder Point-Image Transformer Framework* to fuse image features with geometric point

features, as illustrated in Fig. 2, which maintains reconstruction quality while reducing the attention footprint. Additionally, with an increase in input images, image tokens dominate the attention computation. Inspired by ToME Bolya et al. (2023); Bolya & Hoffman (2023), we merge the image tokens in a fixed ratio $r$ based on the key similarity map calculated from frame-wise attention before global attention, then unmerge them after global attention. Our model effectively fuses multi-image information more efficiently than a straightforward extension of LHM.

The final geometric point features output from the decoder are utilized to regress 3D Gaussian parameters using lightweight multi-layer perceptron (MLP) heads. Subsequently, Linear Blend Skinning (LBS) is employed to animate the canonical avatar into the target pose. To enhance the quality of the animation rendering results, we develop a lightweight, real-time neural rendering network that improves rendering outcomes for novel views and poses based on 3D-aware human animation features.

## 3.2 Encoder-Decoder Point-Image Transformer Framework

To efficiently fuse multi-image features with 3D geometric information, we propose an encoder-decoder architecture based on *Point-Image Transformer blocks (PIT-blocks)*. This framework enables hierarchical feature interaction while alleviating the computational and memory burden associated with dense attention.

We begin by projecting the SMPL-X anchor points in canonical space into a set of geometric tokens and encoding the input images into image tokens, as described in Sec. 3.3. The encoder-decoder network consists of $N_{\text{layer}}$ PIT blocks. In the first $\lfloor N_{\text{layer}}/2 \rfloor$ encoder blocks, we progressively reduce the spatial resolution of the geometric tokens using Grid Pooling Wu et al. (2022). At each layer, the downsampled point tokens perform the attention operation with the image tokens, enabling compact geometric representations enriched with multi-image visual cues.

In the subsequent $\lceil N_{\text{layer}}/2 \rceil$ decoder blocks, we upsample the geometric tokens to restore their original resolution. At each stage, the upsampled tokens are concatenated with the corresponding high-resolution features from the encoder via skip connections. These fused features are further refined by the PIT blocks to reconstruct detailed geometry and view-dependent appearance.

## 3.3 Geometric Point and Images Tokenization

**Geometric Point Tokenization** To incorporate human body priors, we initialize a set of 3D query points $\mathbf{X} = \{\mathbf{x}_i\}_{i=1}^{N_{\text{points}}} \subset \mathbb{R}^3$ by uniformly sampling from the mesh of a canonical SMPL-X pose. Following the design of Point Transformer v3 (PTv3) Wu et al. (2024), we first serialize these points into a structured sequence and then project them into a higher-dimensional feature space using an MLP. Formally, this process is expressed as:

$$X = \text{Serialization}(X),$$
$$\mathbf{T}_{\text{3D}} = \text{MLP}_{\text{proj}}(X) \in \mathbb{R}^{N_{\text{points}} \times C_{\text{point}}}, \tag{2}$$

where $C_{\text{point}}$ denotes the dimensionality of the point tokens.

**Multi-Image Tokenization** To obtain rich image features, we adopt DINOv2 Oquab et al. (2023), a vision transformer pretrained on large-scale in-the-wild datasets, as the image encoder $\mathcal{E}_{\text{Img}}$. Given an input image $I$, we extract a sequence of image tokens as follows:

$$\mathbf{T}_{\text{I}} = \mathcal{E}_{\text{Img}}(I) \in \mathbb{R}^{N_{\text{I}} \times C}, \tag{3}$$

where $N_{\text{I}}$ is the number of image tokens and $C$ is the output feature dimension of the transformer.

## 3.4 Point-Image Transformer Block

After obtaining both geometric and image tokens, we design an efficient *Point-Image Transformer Block* (PIT-block), which comprises three core attention modules to facilitate cross-modal interaction:

**Point-wise Attention** To model self-attention among geometric tokens, we adopt the patch-based point transformer blocks from PTv3 Wu et al. (2024). This design enables cross-patch interactions

via randomized shuffling of point orders, as detailed in the Supplementary Materials:

$$\mathbf{T}_{3D} = \text{PTv3-Block}(\mathbf{T}_{3D}). \tag{4}$$

**Image-wise Attention**  Given the image token sequence $\mathbf{T}_{2D} = \{\mathbf{T}_I^1, \ldots, \mathbf{T}_I^N\} \in \mathbb{R}^{N \times N_I \times C}$, we apply self-attention independently to the tokens of each image. This updates the features within each frame based on its own image tokens:

$$\mathbf{T}_{2D}, \mathbf{K}_{2D} = \text{Self-Attention}(\mathbf{T}_{2D}). \tag{5}$$

where $\mathbf{K}_{2D} \in \mathbb{R}^{NN_I \times C_{attn}}$ represents the key features with $C_{attn}$ channels from $\mathbf{QKV}$ in self-attention block, which are used to compute the similarity map of the image tokens.

**Point-Image Attention**  After obtaining the updated features for both point-wise and frame-wise modalities, we develop a global point-image attention mechanism to fuse the point and multi-image tokens. Our model builds upon the powerful Multimodal-Transformer (MM-Transformer) Esser et al. (2024) to efficiently merge features from different modalities.

To enhance global context representation in the input images, we utilize the class token $\mathbf{T}_{cls}$ extracted from the first frame as learnable global context features. Additionally, to align the dimensions of different modalities, we incorporate projection MLPs into both the input and output layers of the MM-Transformer (MM-T):

$$
\begin{aligned}
\bar{\mathbf{T}}_{3D}, \bar{\mathbf{T}}_{2D} &= \text{MLP}_{proj}(\mathbf{T}_{3D}), \text{Merge}(\text{Flatten}(\mathbf{T}_{2D}), \mathbf{K}_{2D}), \\
\bar{\mathbf{T}}_{3D}, \bar{\mathbf{T}}_{2D} &= \text{MM-T}(\bar{\mathbf{T}}_{3D}, \bar{\mathbf{T}}_{2D} \mid \mathbf{T}_{cls}), \\
\mathbf{T}_{3D}, \mathbf{T}_{2D} &= \text{MLP}_{uproj}(\bar{\mathbf{T}}_{3D}), \text{UnFlatten}(\text{UnMerge}(\bar{\mathbf{T}}_{2D}, \mathbf{K}_{2D}), N).
\end{aligned}
\tag{6}
$$

Specifically, $\text{MLP}_{proj}$ maps point tokens $\mathbf{T}_{3D} \in \mathbb{R}^{\bar{N}_{points} \times C_{point}}$ to $\bar{\mathbf{T}}_{3D} \in \mathbb{R}^{\bar{N}_{points} \times C}$, and $\text{MLP}_{uproj}$ performs the inverse mapping. The Merge and UnMerge operations are adapted from Bolya & Hoffman (2023), with a merge ratio $r = 0.5$, reducing the number of image tokens from $NN_I$ to $NN_I/r$. This design reduces the time complexity of global attention from $\mathcal{O}((N_{points} + N)^2)$ to $\mathcal{O}((\bar{N}_{points} + N/r)^2)$.

### 3.5 3D HUMAN GAUSSIAN PARAMETER PREDICTION

Given the fused point tokens $\mathbf{T}_{3D}$ obtained from the encoder-decoder transformer framework, we predict the parameters of 3D Gaussians in the canonical human space using a lightweight MLP head:

$$
\begin{aligned}
\{\Delta\mathbf{p}_i, \mathbf{r}_i, \mathbf{f}_i, \rho_i, \boldsymbol{\sigma}_i\} &= \text{MLP}_{regress}(\mathbf{T}_{3D}^{(i)}), \\
\mathbf{p}_i &= \mathbf{x}_i + \Delta\mathbf{p}_i, \quad \forall i \in \{1, \ldots, N_{points}\},
\end{aligned}
\tag{7}
$$

where $\Delta\mathbf{p}_i \in \mathbb{R}^3$ denotes the predicted residual offset from the corresponding canonical SMPL-X vertex $\mathbf{x}_i$, and $\mathbf{r}_i, \mathbf{f}_i, \rho_i, \boldsymbol{\sigma}_i$ are the Gaussian orientation, feature vector, opacity, and scale, respectively.

### 3.6 3D-AWARE HUMAN ANIMATION RENDERING

The final output of our model is to drive the reconstructed avatar into view space using the provided human motion control signal, specifically the SMPL-X parameters. Since our Gaussian model is deformed based on SMPL-X, it struggles to accurately represent loose clothing, such as skirts. To enhance the rendering results, we have developed a lightweight, 3D-aware DPT-head for generating final animation outcomes, inspired by LVSM Jin et al. (2025) and HumanRAM Yu et al. (2025).

Unlike HumanRAM, which requires processing through a Transformer with multi-view images each time to produce animation results, our approach allows for real-time generation by constructing animatable avatars in canonical space, enabling conditioning on 3D-aware features (see supplementary material for details).

Table 2. Comparison with sparse-view methods (averaged over public and smartphone video sequences).

| views | InstantAvatar Jiang et al. (2023) | | | | GaussianAvatar Hu et al. (2024) | | | | ExAvatar Moon et al. (2024) | | | | LHM++ M | | | |
| | PSNR | SSIM | LPIPS | Time | PSNR | SSIM | LPIPS | Time | PSNR | SSIM | LPIPS | Time | PSNR | SSIM | LPIPS | Time |
|---|---|---|---|---|---|---|---|---|---|---|---|---|---|---|---|---|
| 2 | 23.337 | 0.944 | 0.091 | 3.6 m | 22.008 | 0.947 | 0.067 | 3.8 m | 26.588 | 0.965 | 0.036 | 8.5 m | **27.835** | **0.970** | **0.018** | **1.08 s** |
| 4 | 23.391 | 0.943 | 0.093 | 6.0 m | 23.382 | 0.952 | 0.051 | 5.6 m | 27.293 | 0.966 | 0.034 | 15 m | **27.940** | **0.971** | **0.018** | **1.40 s** |
| 8 | 23.650 | 0.943 | 0.090 | 10.6 m | 24.106 | 0.956 | 0.048 | 9.0 m | 28.006 | 0.968 | 0.031 | 32 m | **28.147** | **0.971** | **0.017** | **2.09 s** |
| 16 | 24.235 | 0.949 | 0.075 | 18.8 m | 24.398 | 0.957 | 0.047 | 15.7 m | 28.358 | 0.969 | 0.030 | 1.2 h | **28.394** | **0.972** | **0.016** | **3.66 s** |
| 64 | 25.271 | 0.956 | 0.055 | 54.0 m | 25.308 | 0.960 | 0.040 | 1.1 h | **29.587** | **0.971** | **0.027** | 4.4 h | 28.422 | **0.972** | **0.016** | **17.94 s** |

## 3.7 LOSS FUNCTION

Our training strategy integrates photometric supervision from unconstrained video sequences with geometric regularization on Gaussian primitives. This hybrid optimization framework enables the learning of deformable human avatars without the need for explicit 3D ground-truth annotations.

To better capture complex clothing deformations, we adopt a diffused voxel skinning approach as proposed in Lin et al. (2022); Qiu & Chen (2023). Given the predicted 3DGS parameters, we transform the canonical avatar into target view space using voxel-based skinning.

**Photometric Loss** We render the animated Gaussian primitives via differentiable splatting to obtain an RGB image $\hat{I}$ and an alpha mask $\hat{M}$, based on the target camera parameters. Additionally, we generate the neural rendering results $\hat{I}_{NR}$ and $\hat{M}_{NR}$ using a lightweight DPT head. Supervision is applied through the following photometric loss:

$$\mathcal{L}_{photometric} = \lambda_{rgb}\mathcal{L}_{color} + \lambda_{mask}\mathcal{L}_{mask} + \lambda_{per}\mathcal{L}_{lpips}, \qquad (8)$$

where $\mathcal{L}_{color}$ and $\mathcal{L}_{mask}$ represent L1 losses on the RGB and alpha values, corresponding to the 3D Gaussian splatting and neural rendering losses, respectively. The term $\mathcal{L}_{lpips}$ denotes a perceptual loss measures how similar the high-frequency features of neural-rendered and Gaussian Splat–rendered images are to those of the ground-truth images.. The weights are set to $\lambda_{rgb} = 1.0$, $\lambda_{mask} = 0.5$, and $\lambda_{per} = 1.0$.

**Gaussian Regularization Loss** Although the photometric loss offers effective supervision in the target view space, the canonical representation remains under-constrained due to the ill-posed nature of monocular reconstruction. This limitation leads to deformation artifacts when warping the avatar into novel poses. To address this fundamental challenge, we employ Gaussian regularization loss $\mathcal{L}_{reg}$ to improve the learning of unseen parts (see supplementary material for details).

**Overall Loss** The overall training objective combines photometric reconstruction accuracy with geometric regularization:

$$\mathcal{L}_{total} = \mathcal{L}_{photometric} + \mathcal{L}_{reg}. \qquad (9)$$

## 4 EXPERIMENTS

We design three variants of our model with $N_{layer} = 4, 6, 8$ layers of the PI-MT block, corresponding to LHM++ (S), LHM++ (M), and LHM++ (L), respectively. The models comprise approximately 500M, 700M, and 900M training parameters in total. The training details can be found in the supplementary material.

**Evaluation Protocol** We report PSNR, SSIM Wang et al. (2003), and LPIPS Zhang et al. (2018) to assess rendering quality, and measure efficiency with GPU memory usage and inference time on single NVIDIA A100-80G. For our evaluation benchmarks, we establish three datasets: public animation videos, smartphone-captured videos for sparse view reconstruction, and in-the-wild fashion videos for both single-view and sparse view settings. All evaluation datasets will be made publicly available for future research.

### 4.1 COMPARISON WITH EXISTING METHODS

**Animatable Human Reconstruction from Sparse Images** We conduct a comprehensive evaluation of LHM++ by comparing it with three state-of-the-art methods for generating animatable human avatars from casually captured video sequences. We evaluate our model using two types of datasets: one is a public animation benchmarks that includes 23 video sequences from NeuMan Jiang et al.

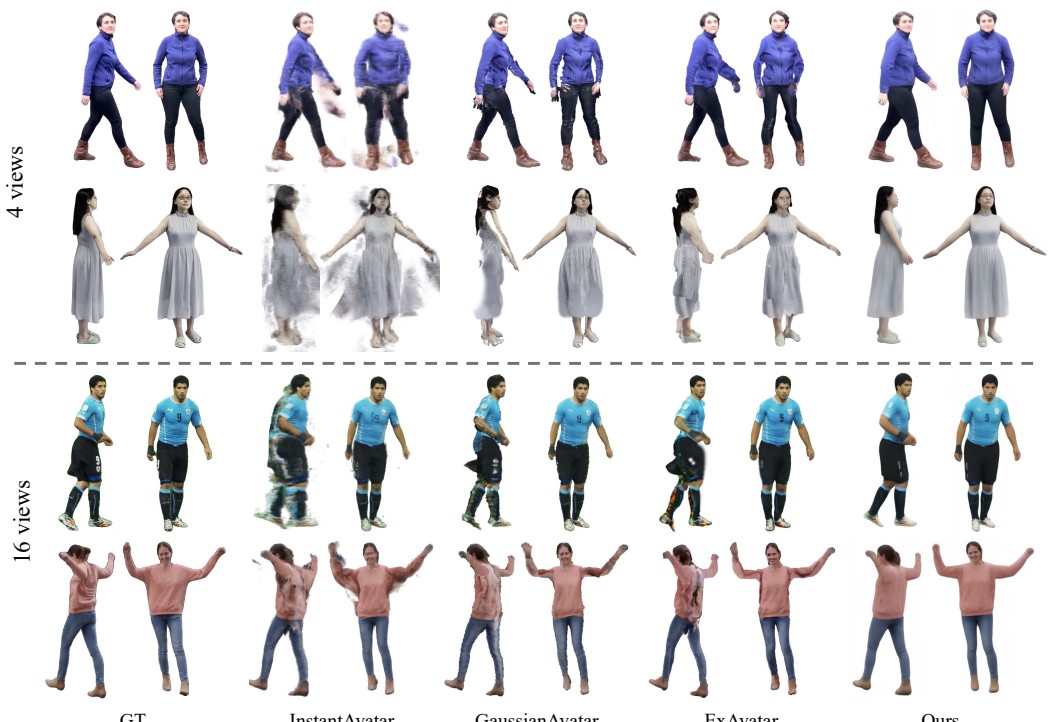

Figure 3. Animatable human reconstruction comparisons from sparse images.

(2022b), REC-MV Qiu & Chen (2023), and Vid2Avatar Guo et al. (2023), while the other comprises 20 smartphone-captured video sequences collected via our smartphones.

Table 2 summarize quantitative experiments evaluating our model against InstantAvatar Jiang et al. (2023), GaussianAvatar Hu et al. (2024), and ExAvatar Moon et al. (2024) on public and smartphone-captured video sequences. Compared to ExAvatar, our method generates animatable avatars within seconds, while ExAvatar requires 15 minutes to over an hour, and achieves comparable quantitative results. Unlike prior approaches that rely on dozens of input images, our model attains higher accuracy with far fewer inputs and continues to improve as more images are provided.

Notably, while the numerical improvements with 16 input views over existing methods on the testing images are slight, our method yields more vivid and realistic novel pose and view animations. As illustrated in Figure 3, sparse input views result in significant reconstruction artifacts when using fitting-based frameworks, including geometric distortions and texture blurring.

**Animatable Human Reconstruction from a Single Image** We evaluate LHM++ against three the state-of-the-art methods of AniGS Qiu et al. (2025b), IDOL Zhuang et al. (2025), and LHM Qiu et al. (2025a). For a fair comparison, we compare LHM with the same parameters, while fine-tuning the pretrained model to accommodate sparse view inputs. Notably, we adopt the 40K query points used in LHM as we observed that when we set the same number of query points as our method, the inference time exceeds 40 seconds for a single-view input.

For our evaluation, we employ 400 in-the-wild video sequences featuring individuals of vari-

Table 3. Comparison with single-image method on pose animation on our in-the-wild fashion dataset. * indicates that we use 40K query points in the official implementation to ensure a fair comparison. Inference speed was measured on a single NVIDIA A100.

| Methods | Input | PSNR ↑ | SSIM ↑ | LPIPS ↓ | Time↓ | Memory ↓ |
|---|---|---|---|---|---|---|
| AniGS Qiu et al. (2025b) | 1 | 17.234 | 0.812 | 0.103 | 15 m | 24.00 GB |
| IDOL Zhuang et al. (2025) | 1 | 17.912 | 0.847 | 0.097 | 1.93 s | 23.12 GB |
| LHM-0.7B* Qiu et al. (2025a) | 1 | 21.233 | 0.877 | 0.078 | 5.73 s | 19.59 GB |
| | 8 | 21.761 | 0.883 | 0.073 | 24.76 s | 25.42 GB |
| | 16 | 21.868 | 0.885 | 0.069 | 49.55 s | 28.77 GB |
| | 64 | 21.852 | 0.885 | 0.069 | 381.48 s | 51.73 GB |
| LHM++ (M) | 1 | 21.518 | 0.882 | 0.068 | 0.91 s | 6.78 GB |
| | 8 | 22.208 | 0.886 | 0.060 | 2.09 s | 7.35 GB |
| | 16 | 22.354 | 0.887 | 0.057 | 3.66 s | 8.48 GB |
| | 64 | 22.363 | 0.887 | 0.057 | 17.94 s | 29.51 GB |

ous age groups, including young men and women, older adults, and children. Figure 4 presents a qualitative comparison of our method with LHM, where our method produces results that are more realistic and detailed compared to LHM. Also, Table 3 and Fig. 5. (b)-(c) show that our integrated framework not only outperforms LHM in terms of quantitative metrics but also achieves an inference

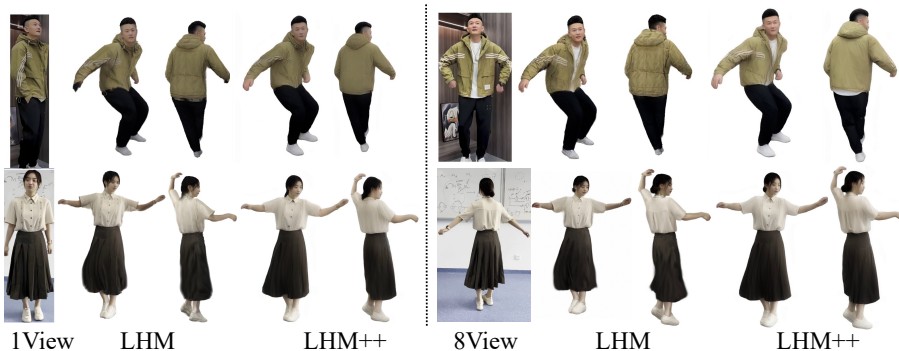

1View  LHM  LHM++  8View  LHM  LHM++

Figure 4. Comparison results of animatable human reconstruction methods with LHM.

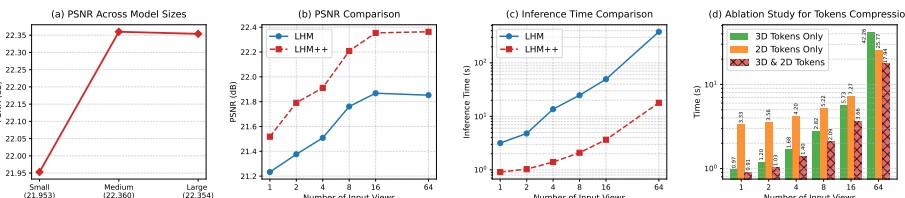

Figure 5. Comparison with LHM based on different input views and the ablation study for Token Compression.

speed that is approximately $5 \sim 10$ times faster. Moreover, the performance of our method improves as the number of input views increases. But, it is worth noting that the gains become marginal with an increasing number of views.

## 4.2 ABLATION STUDY

**Efficiency Analysis** Figure 5. (d) analyzes the effects of compression of image tokens and geometric tokens under different input image numbers. For sparse view inputs, geometric tokens dominate, and their compression significantly affects inference efficiency. As the number of input views increases, the proportion of image tokens gradually rises, making image token compression increasingly effective in accelerating inference and eventually becoming the dominant factor. By jointly optimizing the compression of both token types, our method effectively improves computational efficiency under both sparse and dense view inputs.

Table 4 provides a quantitative analysis of model efficiency and clearly demonstrates the advantages of our pipeline over LHM. LHM requires expensive preprocessing (e.g., foreground parsing and separate extraction of body and head features), so for a fair comparison we report only the inference time for the multi-modality attention stage. Because LHM feeds all 2D and 3D tokens directly into the attention module, its attention computation is extremely costly and does not scale well with the number of 3D tokens. By contrast, our encoder–decoder design reduces redundant 2D and 3D tokens, enabling efficient scaling of the 3D token count. Under the same number of 3D points, our PIT is approximately 70–100× faster than LHM.

Table 4. Analysis of model efficiency. The table shows the time taken by LHM and our method to fuse 2D and 3D tokens (excluding image tokenization and preprocessing) under varying inputs and different numbers of sampling points. Inference speed was measured on a single NVIDIA A100.

| # Points | 1 view | | 4 views | | 8 views | | 16 views | |
| --- | --- | --- | --- | --- | --- | --- | --- | --- |
| | LHM-0.7B | LHM++ (M) | LHM-0.7B | LHM++ (M) | LHM-0.7B | LHM++ (M) | LHM-0.7B | LHM++ (M) |
| 40 K | 5.16 s | **0.55 s** | 9.41 s | **0.74 s** | 15.23 s | **1.01 s** | 32.32 s | **1.74 s** |
| 80 K | 19.47 s | **0.69 s** | 25.73 s | **0.88 s** | 35.36 s | **1.19 s** | 59.25 s | **1.95 s** |
| 160 K | 74.31 s | **0.79 s** | 85.58 s | **1.00 s** | 102.44 s | **1.31 s** | 140.67 s | **2.13 s** |

Moreover, Table 5 illustrates the number of 3D tokens at each encoder layer of the proposed PIT. We use a point-cloud grid-pooling operator to efficiently compress redundant 3D tokens based on their Euclidean positions.

**Model Parameter Scalability** To verify the scalability of our LHM++, we train variant models with increasing parameter numbers by scaling the layer numbers. Figure 5 (a) compares performance across various model capacities. Our experiments indicate that increasing the number of model parameters correlates with improved performance. However, the model size continues to grow and performance improvements tend to saturate. We believe that this limitation is largely influenced by the scale of our training dataset, which is significantly smaller—by an order of magnitude—than the extensive datasets employed in LLMs. Our experiments reveal that while performance shows moderate enhancements from small to medium-sized models, the gains become negligible for larger models. Consequently, we focus on reporting results primarily for the medium-sized model LHM++ (M).

Table 5. 3D token numbers at Encoder layers of PIT.

| Operator | 40K | 80K | 160K |
|---|---|---|---|
| Grid Pooling 1 | 16778 | 22380 | 24377 |
| Point-Image Attention 1 | 16778 | 22380 | 24377 |
| Grid Pooling 2 | 5547 | 6036 | 6441 |
| Point-Image Attention 2 | 5547 | 6036 | 6441 |
| Grid Pooling 3 | 1493 | 1625 | 1686 |
| Point-Image Attention 3 | 1493 | 1625 | 1686 |

**The Effectiveness of Multi-Modality Fusion**

We conduct a comprehensive ablation study to evaluate the efficiency of the proposed PIT module for multi-modality fusion. Table 6 presents comparative experiments between LHM and LHM++, varying the number of sampling points on the in-the-wild fashion dataset with 8 input views. The results show that LHM yields only marginal gains as the number of 3D tokens increases. We attribute this to the need to learn very large attention maps: redundant tokens hinder learning and lead to inefficient fusion. In contrast, the Encoder-decoder PIT structure effectively reduces redundant 3D and 2D tokens and substantially improves the model's fusion performance.

Table 6. Comparison with LHM for multi-modality fusion on our in-the-wild fashion dataset with 8 input views.

| Methods | 40 K | | 80 K | | 160 K | |
|---|---|---|---|---|---|---|
| | PSNR | Time | PSNR | Time | PSNR | Time |
| LHM-0.7B | **21.761** | 24.76 s | 21.803 | 49.78 s | 21.796 | 110.37 s |
| LHM++ (M) | 21.735 | **1.86 s** | **22.124** | **1.98 s** | **22.208** | **2.09 s** |

We also performed an ablation study to examine how the attention modules in the encoder–decoder PIT architecture affect multimodal fusion on our in-the-wild, public, and smartphone video datasets. As shown in Table 7, adding image-wise attention improves per-frame feature learning compared with using only multimodal attention, especially for multi-view inputs, and substantially enhances fusion between 2D and 3D tokens. In addition, point-wise attention which fuses local geometric features using Euclidean distances between points in the point cloud—further improves final performance. These results show that our encoder–decoder PIT is not a simple extension of LHM but an efficient, carefully designed architecture for sparse image inputs.

Table 7. The ablation study for attention modules in the encoder–decoder PIT.

| Methods | In-the-wild Fashion | | Public & Smartphone | |
|---|---|---|---|---|
| | 4 Views | 16 Views | 4 Views | 16 Views |
| LHM++ w/o Image/Point-wise Attention | 21.669 | 21.977 | 27.403 | 27.655 |
| LHM++ w/o Image-wise Attention | 21.735 | 22.136 | 27.656 | 27.927 |
| LHM++ w/o Point-wise Attention | **21.838** | **22.230** | **27.764** | **28.129** |
| LHM++ | **21.957** | **22.354** | **27.940** | **28.394** |

We also perform additional ablation studies investigating model efficiency, dataset scalability, the number of query points, the effects of 3DGS representation for DPT-Head, and the neural renderer (see supplementary materials for details).

## 5 CONCLUSION

We present LHM++, an efficient feed-forward framework for rapid and high-fidelity 3D human avatar reconstruction from one or a few casually captured, pose-free images. Our approach introduces the Encoder-Decoder Point-Image Transformer (PIT), which enables efficient multimodal fusion between geometric point tokens and sparse multi-view image tokens through hierarchical attention. In addition, we introduce a lightweight 3D-aware neural animation renderer to refine the rendering quality of reconstructed avatars. Extensive experiments across synthetic and real-world datasets demonstrate that LHM++ effectively unifies single- and sparse-view reconstruction and supports realistic avatar animation.

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

# APPENDIX

## A    USE OF LLMS

We only utilize Large Language Models (LLMs), such as OpenAI's ChatGPT, and Alibaba's Qwen3-VL, for grammatical corrections, stylistic improvement, and enhancing the clarity and readability of the paper. The scientific narrative, structure, and all claims remain the original work of the authors.

## B    DEMO VIDEO

Please kindly check the Demo Video in the supplementary for animation results of the reconstructed 3D avatar. Our demo consists of three components: render results from varying input views, comparison experiments (against methods such as ExAvatar Moon et al. (2024), LHM Qiu et al. (2025a), and IDOL Zhuang et al. (2025)), and, finally, a series of animation video gallery on both public and in-the-wild dataset.

**Details of Testing Dataset In Demo**    For the comparison experiments with ExAvatar, we utilize 16 input images sampled uniformly from 00000_random and 00069_Dance in Vid2Avatar Guo et al. (2025), anran_self_rotated from REC-MV Qiu & Chen (2023), and Bike from NeuMan Jiang et al. (2022b). In terms of the comparison experiments with feed-forward approaches, we use Anran_Skirt in REC-MV Qiu & Chen (2023), and two in-the-wild monocular videos.

In the animation gallery video, we evaluate our model's generalization capability using four public datasets. For REC-MV Qiu & Chen (2023), we select anran_purple and xiaolin. For NeuMan Jiang et al. (2022b), we use clips from bike, citron, and jogging. For Vid2Avatar Guo et al. (2025), we choose Yuliang, 00000_random, exstrimalik, and Suarez. Finally, for MVHumanNet Xiong et al. (2024), we utilize samples with IDs: 101157, 103383, 103528, and 104316.

**Details of In-the-wild Datasets**    with respect to in-the-wild videos, we utilize web-sourced monocular videos and randomly sample 8 views from the original clips. Regarding the in-the-wild images, we employ FLUX.1 Kontext Batifol et al. (2025) to generate four different view images, using prompts such as "the front view of", "the back view of", "the right view of", and "the left view of".

## C    MORE METHOD DETAILS

### C.1    MORE DETAILS OF THE ENCODER-DECODER POINT-IMAGE TRANSFORMER

**Details of Points Serialization**    To trade the scalability and efficiency of our feed-forward framework, we leverage serialization to transform unstructured SMPL-X anchor points into structured data format. Following Point Transformer v3 (PTv3) Wu et al. (2024), we mixture 4 patterns of serialization, including Z-order, Hibert, Trans Z-order and Trans Hibert, and apply random shuffle to the order of serialization patterns before each PTv3 block.

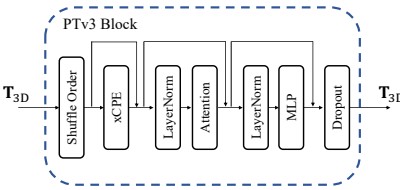

Figure 6. Detailed architecture of PTv3 Block Wu et al. (2024).

**Details of PTv3 Block**    As illustrated in Fig. 6, our point attention architecture integrates the mechanism introduced in Point Transformer v3 Wu et al. (2024). This architecture employs patch-based self-attention to accelerate the forward pass. Furthermore, Grip Pooling is utilized to downsample the point cloud, enhancing the efficiency of point cloud self-attention. Importantly, to facilitate interaction among different patch groups, a shuffling order is implemented.

In our setup, the patch sizes are set to 4096, 2048, and 1024 for the 80K point clouds, decreasing according to the downsampling process. For the 160K point clouds, the patch sizes are configured to 8192, 4096, and 2048.

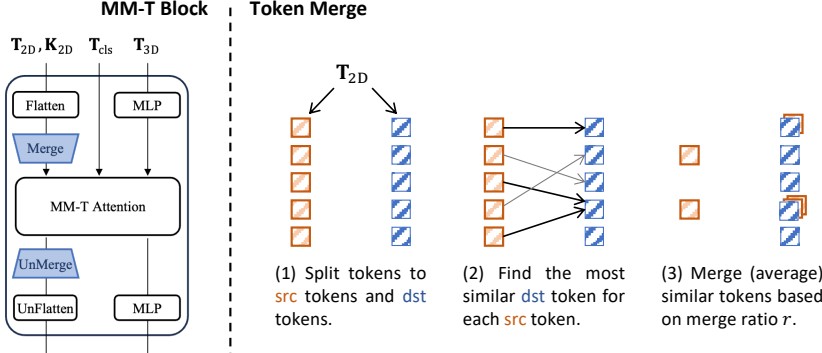

Figure 7. Details of the image token merging process. Given 2D tokens $\mathbf{T}_{2D}$ and key features $\mathbf{K}_{2D}$ from the previous frame-wise attention block, we randomly sample src tokens. We then compute a similarity matrix between the source and target tokens derived from $\mathbf{K}_{2D}$, and greedily merge the most similar tokens from the destination tokens.

**Details of Image Token Merging**  Inspired by ToME Bolya et al. (2023); Bolya & Hoffman (2023), we employ Bipartite Soft Matching to merge redundant image tokens, which helps to reduce inference times during the global attention process. As illustrated in Fig. 7, we construct a bipartite graph using stride sampling as proposed in ToME-SD Bolya & Hoffman (2023). To build the similarity matrix, we utilize the 'Key' tokens from the frame-wise attention mechanism described in the paper. Subsequently, we merge the similar image tokens based on a fixed merge ratio $r$. The merged image tokens are then inputted to compute multi-modal attention alongside the geometric 3D tokens. After the multi-modal attention operation, we unmerge the image tokens back to their original size using a simple feature copying technique, allowing them to proceed to the next PI transformer block.

Surprisingly, we found that reducing a certain proportion of image tokens does not degrade model performance; in fact, it can even lead to a slight improvement. We hypothesize that this improvement results from the removal of redundant tokens, which allows the model to enhance attention weights on key indices within the attention maps. Table 8 presents the details of the ablation study examining the ratio of token merging.

Table 8. **Effects of the reduction ratio used in image token merging.** The metrics are calculated based on 16-view image inputs on in-the-wild fashion videos dataset.

| Methods | PSNR ↑ | SSIM ↑ | LPIPS ↓ | Time ↓ |
|---|---|---|---|---|
| ratio=0.00 | 22.317 | 0.886 | 0.059 | 5.73 s |
| ratio=0.25 | 22.352 | **0.887** | 0.058 | 4.82 s |
| ratio=0.50 | **22.354** | 0.887 | **0.057** | 3.66 s |
| ratio=0.75 | 22.335 | 0.886 | 0.059 | 3.07 s |

## C.2 DETAILS OF 3D-AWARE HUMAN ANIMATION RENDERING

To enhance the rendering results, we have developed a lightweight, 3D-aware DPT-head for generating final animation outcomes, inspired by LVSM Jin et al. (2025) and HumanRAM Yu et al. (2025). Unlike HumanRAM, which requires processing through a Transformer with multi-view images each time to produce animation results—thus slowing down the animation process, our approach allows for real-time generation by constructing animatable avatars in canonical space, enabling conditioning on 3D-aware features.

**Human Animation Feature Rasterization**  As we construct the canonical avatar in canonical space, we can leverage 3D priors to provide fine-grained control information. Specifically, given a reconstructed avatar obtained from Eq. 7 and multi-scale geometric tokens $\mathbf{T}_{3D}$ from Eq. 6, we employ Gaussian Rasterization Qian et al. (2024b), denoted as $F_{GS}$, to render high-dimensional

geometric features into view space, resulting in $\mathbf{I}_{\text{feat}} \in \mathbb{R}^{H \times W \times C_{\text{point}}}$. Notably, as there are no 3D points available to model the background, we utilize learnable background parameters $\mathbf{F}_{\text{back}}$, to represent background colors. The entire process can be formulated as follows:

$$\mathbf{I}_{\text{feat}}^l = \text{F}_{\text{GS}}(\mathbf{T}_{\text{3D}}^l, \mathbf{r}, \mathbf{p}, \rho, \boldsymbol{\sigma}; \mathbf{F}_{\text{back}}). \tag{10}$$

where $\mathbf{T}_{\text{3D}}^l$ represents the geometry features extracted from the $l$-th decoder layer of the PI Transformer framework.

**3D-Aware Neural Animation Rendering**  After obtaining the multi-scale 3D-aware human animation feature map $\mathbf{I}_{\text{feat}}^l$, we employ a lightweight neural network, as proposed in Ranftl et al. (2021), to fuse these multi-scale animation features for novel view and pose rendering:

$$\mathbf{T}_{\text{feat}}^l = \text{Patchify}(\mathbf{I}_{\text{feat}}^l),$$
$$\hat{I}_{\text{NR}}, \hat{M}_{\text{NR}} = \text{DPT-head}(\mathbf{T}_{\text{feat}}^0, \ldots, \mathbf{T}_{\text{feat}}^{N_{\text{layer}}/2}), \tag{11}$$

Here, Patchify refers to the operation that converts the 3D-aware animation features into image tokens using $8 \times 8$ patch windows, $\hat{I}_{\text{NR}}$ and $\hat{M}_{\text{NR}}$ respectively representing the final predicted RGB image and mask results.

## C.3  DETAILS OF GAUSSIAN REGULARIZATION LOSS

Empirically, we observe that using only mask supervision tends to encourage overly large Gaussian scales, especially near object boundaries, which leads to blurred renderings. To counteract this issue, we propose a *Mask Distribution Loss* $\mathcal{L}_{\text{dis}}$, which encourages uniform Gaussian distributions within human masks and sharper boundary representation.

This is achieved by rendering an auxiliary mask $M_{\text{dis}}$ with fixed Gaussian parameters (opacity $\rho = 0.95$, scale $\sigma = 0.002$), and applying a L1 loss between $M_{\text{dis}}$ and the ground-truth human mask.

Furthermore, to reduce ambiguities in canonical space supervision, we adopt two additional geometric regularizers from LHM Qiu et al. (2025a): (1) the *As Spherical As Possible* loss $\mathcal{L}_{\text{ASAP}}$, which promotes isotropy in the 3D Gaussians, and (2) the *As Close As Possible* loss $\mathcal{L}_{\text{ACAP}}$, which preserves spatial coherence among neighboring primitives.

The combined geometric regularization term is defined as:

$$\mathcal{L}_{\text{reg}} = \lambda_{\text{dis}}\mathcal{L}_{\text{dis}} + \lambda_{\text{ASAP}}\mathcal{L}_{\text{ASAP}} + \lambda_{\text{ACAP}}\mathcal{L}_{\text{ACAP}}, \tag{12}$$

with empirically chosen weights: $\lambda_{\text{dis}} = 0.5$, $\lambda_{\text{ASAP}} = 20$, and $\lambda_{\text{ACAP}} = 5$.

## C.4  IMPLEMENTATION DETAILS

We design three variants of our model with $N_{\text{layer}} = 4, 6, 8$ layers of the PI-MT block, corresponding to LHM++ (S), LHM++ (M), and LHM++ (L), respectively. The models comprise approximately 500M, 700M, and 900M training parameters in total. To ensure stable training, we first train the single-view variant, LHM++, by minimizing the training loss using the AdamW optimizer for 65 K iterations without incorporating the parameters from the neural rendering component. Following this, we randomly sample between 1 and 16 frames from a randomly selected training video as input to establish a pose-free setting for any image input during each batch. We continue training the model by minimizing the training loss with the AdamW optimizer for an additional 41 K iterations. Additionally, we apply gradient norm clipping with a threshold of 0.1. We utilize bfloat16 precision and gradient checkpointing to improve GPU memory usage and computational efficiency. A cosine learning rate scheduler is employed, with a peak learning rate of 1e-4 and a warm-up period of 3,000 iterations. Input images are resized to have a maximum dimension of 1024 pixels. The training is conducted on 32 A100 GPUs over a duration of seven days.

**Training Dataset**  For our model training, we utilize approximately 300,000 in-the-wild video sequences collected from public video repositories, along with over 5,173 3D public synthetic static

human scans sourced from 2K2K Han et al. (2023), Human4DiT Shao et al. (2024), and RenderPeople. Specifically, we employ a sampling ratio of 19:1 to draw training batches from the in-the-wild and synthetic datasets to balance generalization and view-consistency. To address view bias in the video data, we sample from a diverse range of perspectives as uniformly as possible, guided by the estimated global orientation of SMPL-X.

# D EXPERIMENTS

## D.1 DETAISL OF PUBLIC ANIMATION BENCHMARKS

We collect 23 public videos to form our evaluation benchmark. Specifically, we selected anran_dance_self_rotated, anran_purple, anran_skirt, lingteng, self-rotate-leyang, and xiaolin from Rec-MV Qiu & Chen (2023). Additionally, we sampled bike, citron, parkinglot, jogging, lab, and seattle from NeuMan Jiang et al. (2022b). Furthermore, we included 00000_random, 00020_Dance, 00069_Dance, exstrimalik, helge, lam, Nadia, phonecall, roger, truman, and Yuliang from Vid2Avatar Guo et al. (2023; 2025).

## D.2 QUANTITATIVE RESULTS

Table 2 of the main paper summarizes the averaged results on the public and smartphone-captured video sequences. Table 9 and Table 10 show the detailed results for each dataset.

Table 9. Comparison experiments with sparse-view input methods on public animation benchmark.

| views | InstantAvatar Jiang et al. (2023) | | | | GaussianAvatar Hu et al. (2024) | | | | ExAvatar Moon et al. (2024) | | | | LHM++ M | | | |
|---|---|---|---|---|---|---|---|---|---|---|---|---|---|---|---|---|
| | PSNR | SSIM | LPIPS | Time | PSNR | SSIM | LPIPS | Time | PSNR | SSIM | LPIPS | Time | PSNR | SSIM | LPIPS | Time |
| 2 | 21.718 | 0.929 | 0.125 | 3.6 m | 21.049 | 0.937 | 0.086 | 3.8 m | 26.371 | 0.959 | 0.045 | 8.5 m | **26.900** | **0.964** | **0.022** | **1.08 s** |
| 4 | 21.782 | 0.926 | 0.126 | 6.0 m | 22.614 | 0.943 | 0.065 | 5.6 m | 26.906 | 0.960 | 0.044 | 15 m | **27.031** | **0.965** | **0.022** | **1.40 s** |
| 8 | 22.308 | 0.928 | 0.120 | 10.6 m | 23.238 | 0.947 | 0.061 | 9.0 m | 27.319 | 0.962 | 0.040 | 32 m | **27.346** | **0.965** | **0.021** | **2.09 s** |
| 16 | 23.026 | 0.936 | 0.101 | 18.8 m | 23.427 | 0.949 | 0.061 | 15.7 m | 27.456 | 0.962 | 0.040 | 1.2 h | **27.521** | **0.966** | **0.020** | **3.66 s** |
| 64 | 23.561 | 0.944 | 0.077 | 54.0 m | 23.574 | 0.949 | 0.055 | 1.1 h | **27.896** | 0.962 | 0.037 | 4.4 h | 27.557 | **0.966** | **0.020** | **17.94 s** |

Table 10. Comparison experiments with sparse-view input methods on smartphone-captured videos.

| views | InstantAvatar Jiang et al. (2023) | | | | GaussianAvatar Hu et al. (2024) | | | | ExAvatar Moon et al. (2024) | | | | LHM++ M | | | |
|---|---|---|---|---|---|---|---|---|---|---|---|---|---|---|---|---|
| | PSNR | SSIM | LPIPS | Time | PSNR | SSIM | LPIPS | Time | PSNR | SSIM | LPIPS | Time | PSNR | SSIM | LPIPS | Time |
| 2 | 24.340 | 0.954 | 0.070 | 3.6 m | 22.602 | 0.953 | 0.056 | 3.8 m | 26.723 | 0.969 | 0.031 | 8.5 m | **28.413** | **0.974** | **0.016** | **1.08 s** |
| 4 | 24.387 | 0.953 | 0.072 | 6.0 m | 23.857 | 0.957 | 0.043 | 5.6 m | 27.532 | 0.970 | 0.028 | 15 m | **28.502** | **0.974** | **0.016** | **1.40 s** |
| 8 | 24.481 | 0.952 | 0.072 | 10.6 m | 24.644 | 0.961 | 0.040 | 9.0 m | 28.431 | 0.972 | 0.026 | 32 m | **28.643** | **0.974** | **0.015** | **2.09 s** |
| 16 | 24.984 | 0.957 | 0.059 | 18.8 m | 24.999 | 0.962 | 0.039 | 15.7 m | 28.916 | 0.973 | 0.024 | 1.2 h | **28.934** | **0.975** | **0.014** | **3.66 s** |
| 64 | 26.329 | 0.964 | 0.042 | 54.0 m | 26.381 | 0.967 | 0.030 | 1.1 h | **30.634** | **0.978** | 0.020 | 4.4 h | 28.958 | 0.975 | **0.013** | **17.94 s** |

## D.3 DETAILS OF 3D-AWARE HUMAN ANIMATION NEURAL RENDERING TIME

To enhance the rendering results, we have developed a lightweight neural network that enhance the overall rendering process. By reconstructing a detailed avatar in canonical space with high-dimensional semantic features, and leveraging the 3D-aware human prior, our model is capable of rendering final results in real-time. Table 11 clearly illustrates the rendering times achieved by our 3D-aware human animation neural renderer.

Table 11. Rendering Times of the 3D-Aware Human Animation Neural Renderer on a Single NVIDIA A100-80G

| Resolution | Time (s) ↓ | FPS ↑ |
|---|---|---|
| $592 \times 592$ | 0.013 | 78 |
| $880 \times 880$ | 0.027 | 37 |
| $1176 \times 1176$ | 0.048 | 21 |

### D.4 Discussion on General image-to-3D generation

We conducted comparative experiments against general image-to-3D methods. Recent cutting-edge image-to-3D large models include Rodin Zhang et al. (2024a) and Hunyuan3D-2.5 Team (2025). For our experiments, we chose Hunyuan3D-2.5 as the baseline because of its state-of-the-art performance. After reconstructing a 3D model from the input-view avatar, we use Mixamo to auto-rig the models, converting any input image into an animatable avatar.

As shown in Fig. 8, rigging a 3D digital human places very strict requirements on the input pose. While Hunyuan3D-2.5 can recover the overall structure of the input image, it often fails to produce a riggable geometry. This is evident in the "Reset" panel, where points on the arms and torso are hard to distinguish, making the reconstructed avatar difficult to animate. In addition, the generated textures frequently appear unnatural and contain artifacts or unrealistic details. In contrast, LHM++ produces natural, high-fidelity avatars in canonical space.

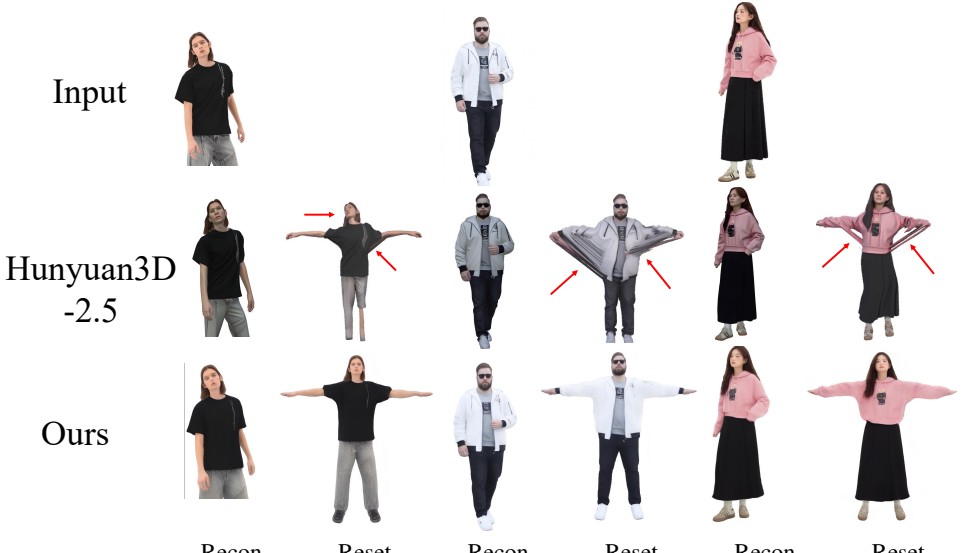

Figure 8. **Comparison with Hunyuan3D-2.5**. We use Hunyuan3D-2.5 to reconstruct 3D models in the input-view space. We then auto-rig the models using Mixamo and reset the target-view pose into the canonical space. "Recon" denotes reconstruction in the input space, while "Reset" denotes animating the avatar into a predefined canonical pose.

### D.5 Failure Case

Figure 9 shows failure cases from our model. It has difficulty reconstructing avatars from images of people in extreme or challenging poses, and it also struggles to animate avatars wearing skirts during large lower-body movements.

### D.6 More Ablation Study

**Model Efficiency** Table 12 presents quantitative results that substantiate the efficiency of our model, tested on NVIDIA A100-80G hardware. The table clearly illustrates that our novel framework significantly outperforms the original LHM architecture, achieving notably reduced training times and lower memory consumption. Also, with the same query points input, the inference time of LHM++ is roughly 55× faster than LHM. Notably, the reported inference times include additional image tokenization time. Therefore, these values differ from those in Table 4

**Details of Dataset Scalability** To assess the scalability of our dataset, we perform controlled experiments utilizing stratified random subsets of 10K and 100K from the original training dataset of 300K videos. Table 13 demonstrates that relying solely on the synthetic dataset leads to poor model generalization. In contrast, incorporating an in-the-wild dataset significantly improves the

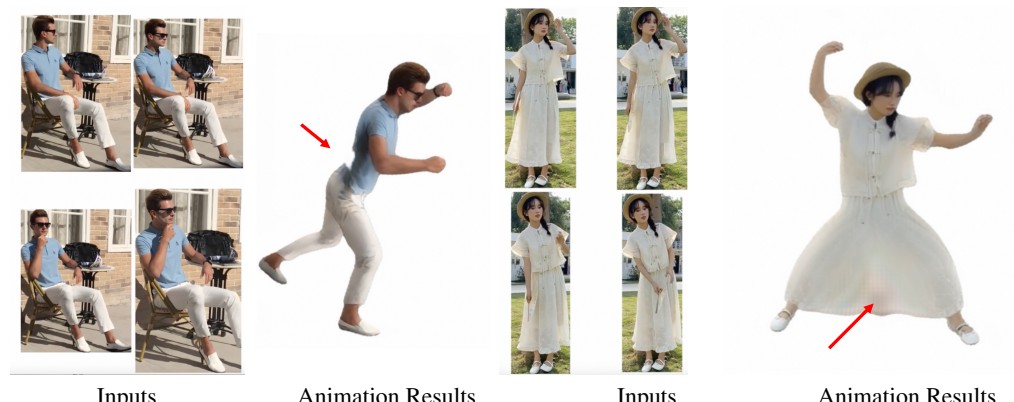

| Inputs | Animation Results | Inputs | Animation Results |

Figure 9. **Failure Cases.** our model struggles to reconstruct avatars from images of people in extreme or challenging poses, and it also struggles to animate avatars wearing skirts during large-amplitude lower-body movements.

Table 12. **Model Efficiency.** We evaluate the training and inference efficiency of various backbones. The batch size is set to 2, the number of input views is equal to 8. '# Points' refers to the number of geometric points, and 'Time' denotes the duration of a single iteration during both training and inference. The 'OOM' indicates an out-of-memory error. Inference speed was measured on a single NVIDIA A100.

| Methods | Params. | # Points | Training | | Inference | |
|---|---|---|---|---|---|---|
| | | | Time | Memory | Time | Memory |
| LHM-0.7B* Qiu et al. (2025a) | 700 M | 40 K | 81.22 s | 80.3 GB | 24.76 s | 25.42 GB |
| | 700 M | 160 K | - | OOM | 110.3 s | 36.71 GB |
| LHM++ (S) | 500 M | 160 K | **5.79 s** | **43.5 GB** | **1.58 s** | **6.98 GB** |
| LHM++ (M) | 700 M | 160 K | **6.35 s** | **48.7 GB** | **2.09 s** | **7.35 GB** |
| LHM++ (L) | 900 M | 160 K | 7.25 s | 52.6 GB | 2.88 s | 8.21 GB |

model's robustness and performance during real-world evaluations. Furthermore, increasing the size of the dataset siginifcantly improves model results, although the gains in performance tend to diminish with larger dataset sizes. Figure 10 illustrates the findings of our ablation study regarding dataset scalability.

Table 13. **Ablation study on the dataset scalability.** The metrics are calculated based on 16-view image inputs on in-the-wild fashion videos dataset.

| Methods | PSNR ↑ | SSIM ↑ | LPIPS ↓ |
|---|---|---|---|
| Synthetic Data | 20.549 | 0.858 | 0.086 |
| 10K Videos | 21.526 | 0.870 | 0.073 |
| 100K Videos | 22.051 | 0.085 | 0.062 |
| 300K Videos + Synthetic Data | **22.354** | **0.887** | **0.057** |

**Number of Query Points**  Table 14 shows an ablation study analyzing the effect of varying the number of query points on public video datasets. As the number of query points increases from 40K to 160K, our model demonstrates improvements in PSNR, SSIM, and LPIPS by 0.564, 0.006, 0.01, respectively. However, when the number of query points is increased further from 160K to 320K, we observe a slight decrease in performance.

**3D-Aware Human Neural Renderer**  To improve the quality of the rendering outcomes, we have developed a lightweight 3D-aware human neural renderer, specifically the 3D-aware DPT-head, aimed at refine final animation rendering results. Table 14 presents a quantitative ablation study comparing the Gaussian Splat renderer with the 3D-aware human neural renderer. The experiments indicate, when using the same renderer (Gaussian Renderer), the proposed framework not only outperforms LHM in photometric performance but also achieves significant improvements in inference time.

Synthetic    10K    100K    300K    GT

Figure 10. Ablation study on dataset scalability.

Table 14. The ablation study examining the impact of 3D geometric point numbers, both with and without the 3D-aware neural renderer. The metrics are computed using 16-view image inputs from an in-the-wild fashion video dataset.

| Methods | # Points | PSNR ↑ | SSIM ↑ | LPIPS ↓ | Time↓ | Memory ↓ |
|---|---|---|---|---|---|---|
| LHM++ (M) | 40 K | 21.790 | 0.881 | 0.067 | 3.04 s | 7.76 GB |
| LHM++ (M) | 80 K | 22.157 | 0.884 | 0.060 | 3.27 s | 8.02 GB |
| LHM++ (M) | 160 K | **22.354** | 0.887 | **0.057** | 3.66 s | 8.48 GB |
| LHM++ (M) | 320 K | 22.342 | **0.887** | 0.057 | 4.23 s | 8.89 GB |
| LHM-0.7B | 40 K | 21.868 | 0.885 | 0.069 | 49.55 s | 28.77 GB |
| Ours w/o Neural Renderer | 160 K | 22.305 | 0.886 | 0.066 | 3.64 s | 8.29 GB |
| Ours w/ Neural Renderer | 160 K | **22.354** | **0.887** | **0.057** | 3.66 s | 8.48 GB |

Figure 11 illustrates the qualitative results of the ablation study for 3D-aware neural renderer. Since our LHM++ directly learns a positional residual from the SMPL-X template, challenges remain in rendering loose-fitting clothing, despite employing mask distribution loss. The proposed neural renderer not only effectively addresses this issue, but also significantly enhance the overall rendering results.

**Mask Distribution Loss**  Gaussian models tend to learn large-scale Gaussian primitives rather than large step offsets. To address this issue, we introduce a mask distribution regularization loss that encourages Gaussian primitives to learn offsets instead of concentrating on large-scale axes. As illustrated in Fig. 12, employing this loss enables the mean position of the Gaussian primitives to be distributed more evenly across the ground truth mask area, thereby preventing the model from learning large-scale axes of the Gaussian primitives.

**Effects of the 3DGS representation for the DPT head**  We conducted an ablation in which we deformed only the points sampled from SMPL-X (modifying their positions) and rendered features

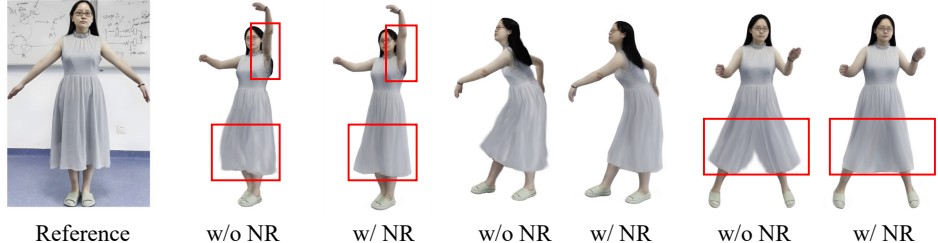

Reference    w/o NR    w/ NR    w/o NR    w/ NR    w/o NR    w/ NR

Figure 11. The ablation study for 3D-aware human neural renderer. 'NR' is the abbreviation for '3D-aware neural renderer'. In this study, we used 8 images as input.. The red boxes highlight the visual difference mentioned above.

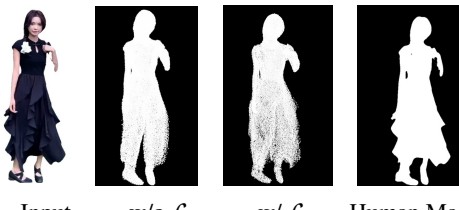

| Input | w/o $\mathcal{L}_{dis}$ | w/ $\mathcal{L}_{dis}$ | Human Mask |

Figure 12. The ablation study for mask distribution loss.

via geometric rasterization implemented with PyTorch3D, followed by rendering with the DPT head — i.e., omitting the 3DGS representation used in our full pipeline.

Apart from the 3D representation, all training settings were identical to those of LHM++. As the Fig. 13 shows, relying solely on point offsets without an explicit 3DGS representation causes the DPT head to struggle to reproduce the input-image context, producing high-frequency artifacts and blurring; the pipeline also has difficulty converging. In contrast, using an explicit 3DGS representation enables the DPT head to converge faster and substantially improves overall visual quality.

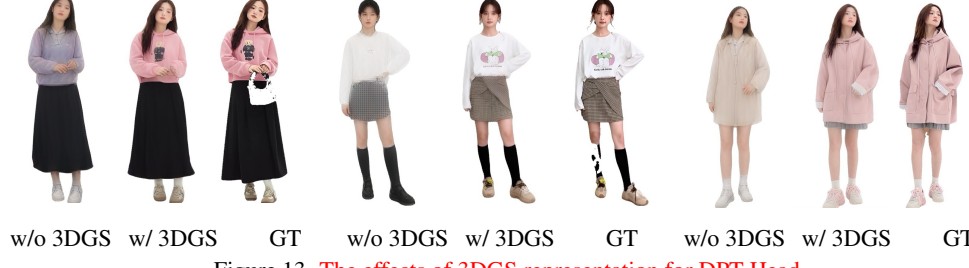

| w/o 3DGS | w/ 3DGS | GT | w/o 3DGS | w/ 3DGS | GT | w/o 3DGS | w/ 3DGS | GT |

Figure 13. The effects of 3DGS representation for DPT-Head.

### D.7 MORE RESULTS

**Human Reconstruction in Canonical Space** Figure 14 demonstrates human reconstructions in canonical space from sparse views, showing that our model can reconstruct reasonable, high-quality avatars from sparse images input.

**More Animation Rendering Results** Both Fig. 15 and Fig. 16 present more animation results using eight input images. Our method enables high-fidelity reconstruction and animation of human avatars through the efficient Point-Image Transformer architecture, thereby demonstrating robust generalization capabilities and practical effectiveness.

## E LIMITATIONS AND FUTURE WORK

A primary limitation of our approach is its dependence on the SMPL-X template mesh to initialize geometric tokens. While this provides a strong structural prior, it can limit reconstruction fidelity for subjects wearing loose or non-body-conforming garments, such as dresses, that diverge significantly from the SMPL-X topology. Furthermore, due to the limited diversity of large-motion poses in our training datasets, the model's performance may decline when faced with unseen or extreme poses. Additionally, since our model is primarily designed for sparse view input, we find that as the number of input views exceeds 16, the improvement in performance becomes minimal. In future work, we plan to investigate more flexible garment representations and explore pose-independent anchor structures to better capture complex clothing dynamics, enhancing generalization to diverse motions. We also aim to design a framework that is more suitable for dense view inputs.

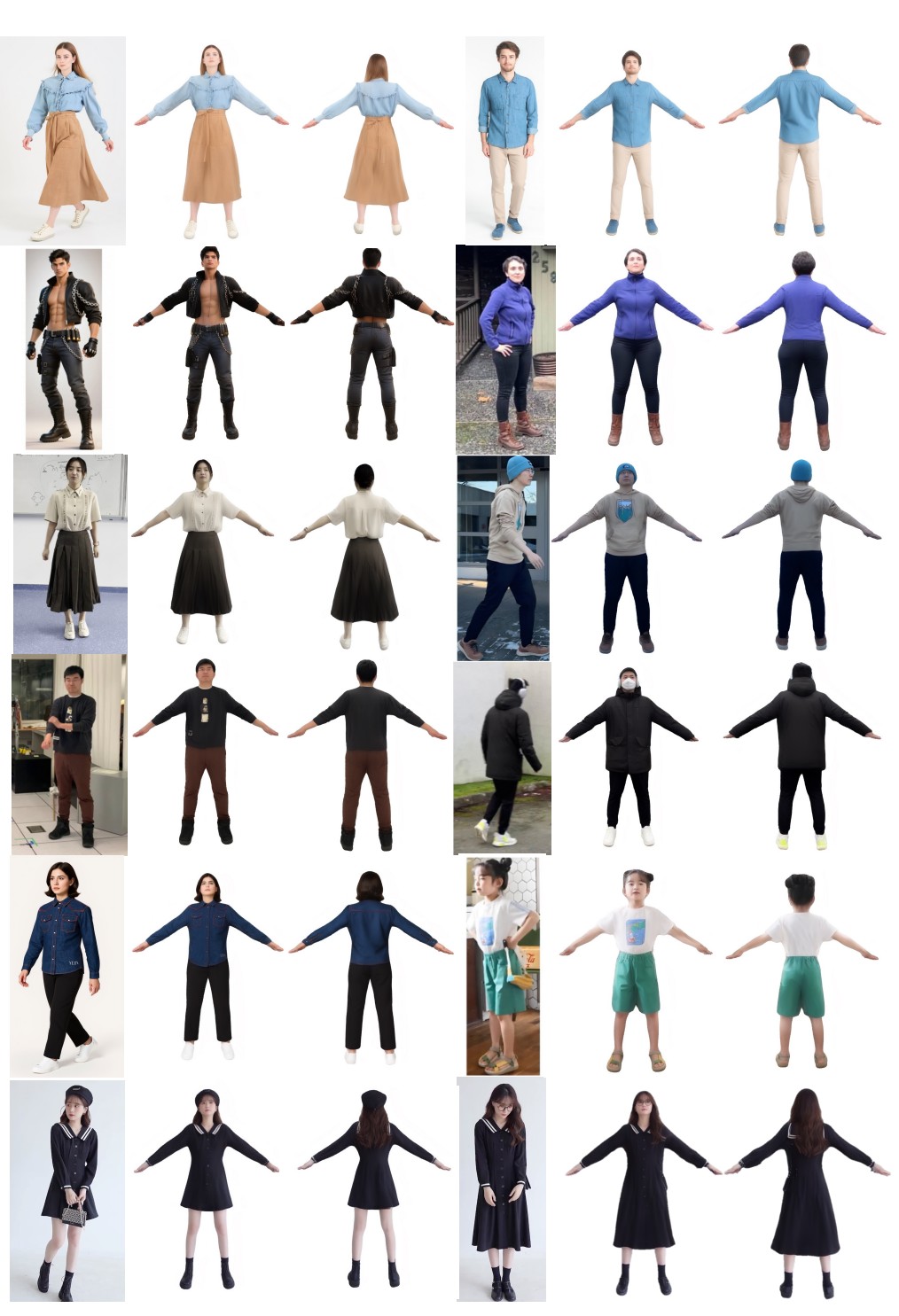

Reference          Reconstruction          Reference          Reconstruction

Figure 14. Human reconstruction in canonical space from 8 input images. The reference image is one of the inputs.

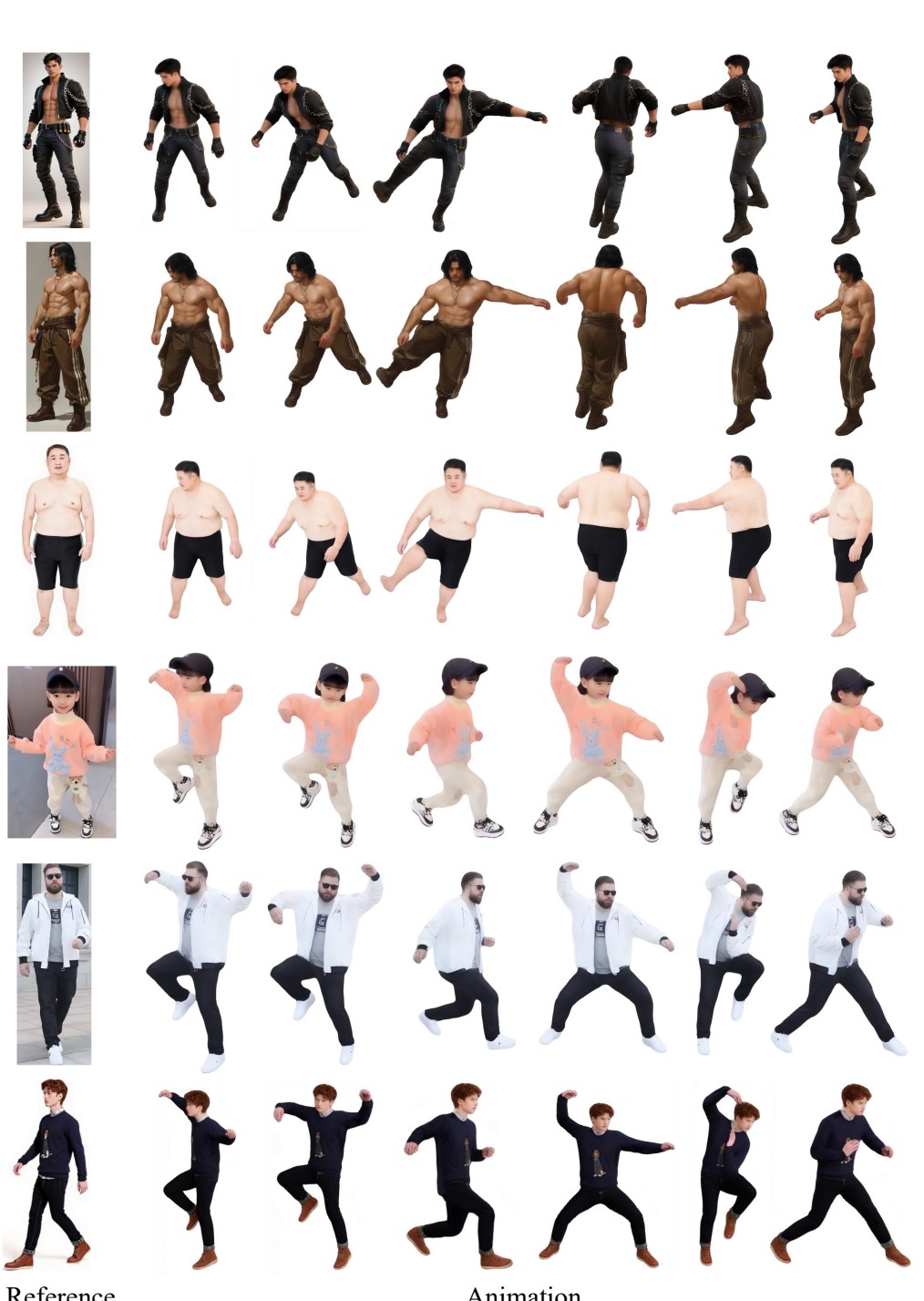

Reference                                    Animation

Figure 15. More animation results of avatars created with 8-image inputs (Part I). **Reference** image is one of the input images.

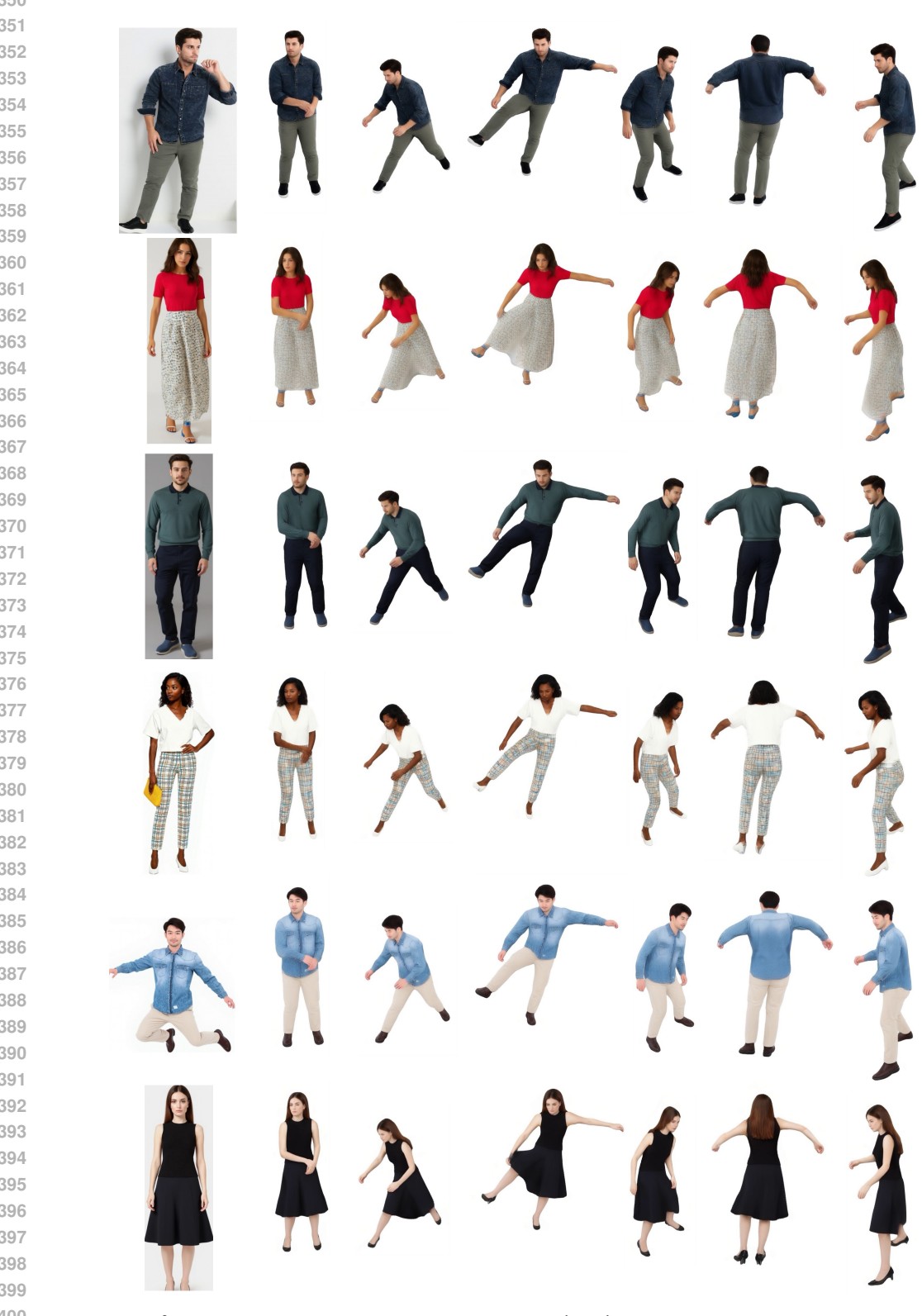

Reference                  Animation

Figure 16. More animation results of avatars created with 8-image inputs (Part II). **Reference** image is one of the input images.

# F UPDATED EXPERIMENTS

## F.1 ABLATION STUDY FOR IMAGE TOKENS MERGE

To further analyze the 'merge' and 'unmerge' operator in global image-point attention, we conducted both qualitative and quantitative experiments that demonstrate how performance varies with different merge ratios and numbers of views.

Figure 17 illustrates qualitative results showing how visual quality changes with different token compression ratios across sparse-view inputs. The figure indicates that for $1 \sim 4$ views a high merge ratio $r > 0.5$ significantly degrades visual quality, whereas for $8 \sim 16$ views, larger merge ratios $r \in [0.5, 0.75]$ do not obviously reduce performance. We attribute this to the increased redundancy of tokens as the number of views grows, and the our model de facto does not require all tokens.

Moreover, Table 15 presents a quantitative ablation of image-token merging. The table reports PSNR averaged on both public and smartphone video sequences. It shows that applying token merging with moderate ratios $r \in [0.25, 0.75]$ for sparse-view inputs $4 \sim 16$ yields metrics that are similar to, or slightly better than, the no-merge baseline $r = 0$. This results is consistent with findings from previous work in image generation Bolya & Hoffman (2023).

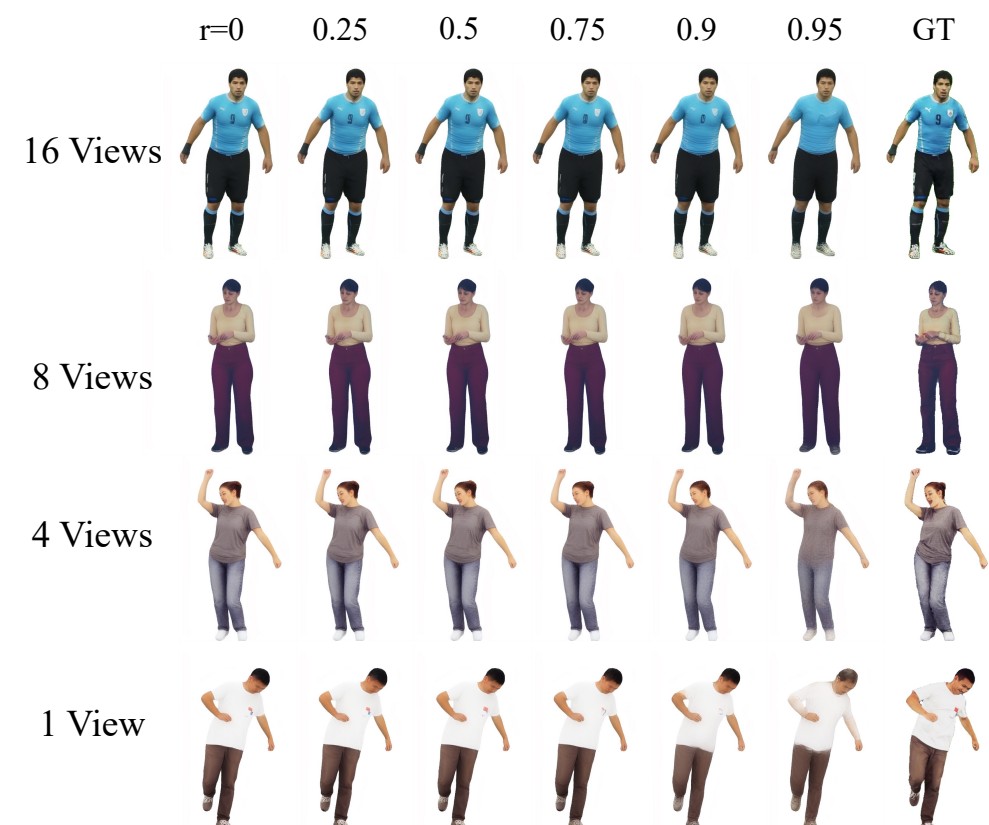

Figure 17. Ablation study of image-token merging. 'r' denotes the merge ratio ($r = 0$ means no merging). Larger 'r' values correspond to greater compression of image tokens. Please zoom in for better view.

Table 15. **Ablation study of image-token merging.** The PSNR metrics are averages computed over public and smartphone video sequences.

| Input Views | $r = 0.00$ | $r = 0.25$ | $r = 0.50$ | $r = 0.75$ | $r = 0.90$ | $r = 0.95$ |
|---|---|---|---|---|---|---|
| view = 1 | **27.762** | 27.735 | 27.733 | 27.644 | 27.525 | 27.238 |
| view = 4 | 27.942 | **27.942** | 27.940 | 27.937 | 27.893 | 27.662 |
| view = 8 | 28.140 | **28.153** | 28.147 | 28.135 | 27.903 | 27.735 |
| view = 16 | 28.391 | **28.394** | **28.394** | 28.391 | 28.358 | 28.206 |

