# OpenReview forum: "LHM++: An Efficient Large Human Reconstruction Model for Pose-free Images to 3D"
_ICLR.cc/2026/Conference — Submitted to ICLR 2026_

### Official Review · Reviewer_nJbr · 2025-10-22

**Soundness:** 3
**Presentation:** 3
**Contribution:** 2
**Rating:** 4
**Confidence:** 4

**Summary:**

The paper presents an advanced version of LHM that accepts multiple images and produces high-quality, animatable 3D avatars with a large feed-forward model. The authors propose an Encoder–Decoder Point–Image Transformer that fuses 3D points with image features to handle multiple images efficiently. The fused tokens are decoded into 3D Gaussian splats and rendered with a lightweight 3D-aware neural renderer for real-time animation.

**Strengths:**

- Paper is well-written and easy to understand.
- Extensive ablation studies and visualizations are provided in the main paper and supplementary materials.
- The proposed Encoder–Decoder Point–Image Transformer reduces inference time compared to LHM, especially as the number of images increases, while also improving performance.

**Weaknesses:**

- The main concern is that the network is designed for sparse views. Why is this design choice necessary? Although the authors extend LHM to multi-image input, the paper shows limited gains beyond 16 images (e.g., Table 2/5 at 64 views). This raises the question of whether the multi-image extension offers meaningful benefits at higher view counts. In particular, the paper notes that “the gains become marginal with an increasing number of views” (Line 431), which is counter-intuitive if more views should provide more information. Please clarify.

- The overall framework feels close to LHM: both use multimodal transformers to fuse 3D geometry with image features inside the network. The technical contribution beyond LHM on model design is not entirely clear from the framework description.

- In Model Design, the paper highlights LHM’s quadratic attention complexity O(N_points + N)^2, but in Point–Image Attention the complexity remains quadratic after token merging, while N reduced to N/r. This suggests limited improvement as N grows large. A more detailed complexity analysis would strengthen the efficiency claim.

- There is no visualization of avatar on canonical pose. It would be nice if the canonical pose visualization is included.

- Missing recent references (recommend adding and, if possible, comparing in Table 1):

    [1] Kocabas, Muhammed, et al. "Hugs: Human gaussian splats." CVPR 2024. → Monocular video based reconstruction

    [2] Shin, Jisu, et al. "Canonicalfusion: Generating drivable 3d human avatars from multiple images." ECCV 2024. → Monocular video based reconstruction

    [3] Liao, Tingting, et al. "High-fidelity clothed avatar reconstruction from a single image." CVPR 2023. → Single image based reconstruction

    [4] Moreau, Arthur, et al. "Human gaussian splatting: Real-time rendering of animatable avatars." CVPR 2024. → Multi-view video based reconstruction

    [5] Wang, Rong, et al. "FRESA: Feedforward Reconstruction of Personalized Skinned Avatars from Few Images." CVPR 2025 → Few image based reconstruction

I will reconsider the score when all the concerns are handled well.

**Questions:**

- What is the main difference between LHM and this paper’s framework beyond the point–image attention details?

- Why doesn’t performance improve after 16 images? Is this due to training distribution, or difficulty handling deformations?

- What are the common failure cases of the framework (e.g., loose garments, extreme poses, or heavy occlusions)? A small failure case visualization would be informative.

---

> ### Author Response · Authors · 2025-11-21
> **Official Comment by Authors-A**
>
> ***To Reviewer #nJbr:***
>
> We thank the reviewer for the appreciation of the **clearly written paper**, the **extensive experimental validation and visualization,** and **effienency** of LHM++. We address the weaknesses and questions point-by-point below.
>
> ---
>
> ***W1-1: Why sparse views design choice necessary?***
>
> Single-view image input inherently lacks sufficient geometric and appearance information. For instance, when only a front-view is provided, the model has no access to information about the back side of the subject. In such cases, the model must largely **hallucinate** the unseen regions, often resulting in **blurry** or **inaccurate** reconstructions (**As shown in our supplementary video, from the 4th second to the 16th second.**).
>
>
>
> ***W1-2: Why the gain of dense view input is marginal?***
>
> We would like to clarify that our method is **feed‑forward** and uses **self‑attention** to **implicitly** aggregate complementary information across views during inference. Unlike **optimization‑based** approaches (e.g., ExAvatar), which iteratively fit a model to each input, our approach produces the avatar in a single forward pass. In other words, the model acts as a **generative** **predictor** that maps the available views to an avatar appearance.
>
>
>
> **Empirically, 8–16 well‑distributed views** already provide **near‑complete** coverage of the subject, so additional views typically add little new information. Consequently, performance saturates as view density increases, and very dense multi‑view inputs rarely yield substantial gains.

---

> ### Author Response · Authors · 2025-11-21
> **Official Comment by Authors-B**
>
> ### ***W2: What is the technique contribution of LHM++ compared to LHM?***
>
>
>
> We would be happy to provide a more detailed explanation of LHM++'s technical contributions.
>
>
>
> The primary technical contribution is the proposed encoder–decoder  Point–Image Transformer architecture, which significantly reduces  inference time while delivering better performance than LHM on  sparse-view inputs. We also introduce a lightweight 3D-aware  human-animation neural renderer to improve visual quality in  **under-constrained regions**.
>
>
>
> Regarding the architectural contribution, we clarify that the PIT **is not a simple extension** of LHM but an **efficient, carefully designed**  architecture for sparse image inputs that delivers **substantial** speedups  and improved results.
>
>
>
> First, we highlight the performance gains of our PIT architecutre. We obsever the number of 3D points is critically important for model performance: a larger point count enables the model to capture **finer geometric** details. However, this **does not imply that increasing the number of points invariably leads to better learning outcomes**. Under conditions with more point input, the **key** to further performance gains lies in **effectively fusing sparse image inputs information** with the 3D point representations.
>
> ---
>
>
>
> #### ***Contribution for Performance Gain***
>
> To support the opinion, we ran comparative experiments with LHM while varying the number of 3D tokens using an 8-view input. the results are reported in the following table (**[Tab.6 Lines 501-507]** in the revision**) .**
>
>
>
> | Methods   | 40 K PSNR  | 40 K Time  | 80 K PSNR  | 80 K Time  | 160 K PSNR | 160 K Time |
> | --------- | ---------- | ---------- | ---------- | ---------- | ---------- | ---------- |
> | LHM-0.7B  | **21.761** | 24.76 s    | 21.803     | 49.78 s    | 21.796     | 110.37 s   |
> | LHM++ (M) | 21.735     | **1.86 s** | **22.124** | **1.98 s** | **22.208** | **2.09 s** |
>
> The results show that LHM yields only marginal gains as the number of 3D tokens increases. We attribute this to the need to learn **extremely large attention maps**. For example, attention matrices can contain **200K * 200K** (160K for 3D tokens, 40K for 2D tokens in LHM setting) configurations with 160K 3D tokens and 8 views image tokens. Such redundant tokens **hinder** learning and make **fusion inefficient**. In contrast, the Encoder-Decoder PIT architecture **reduces redundant 3D and 2D tokens** and **substantially improves fusion efficiency and overall performance**.
>
>
>
> To further analyze the improvements introduced by PIT, we conducted an ablation study on each attention module. As shown in the following table (**Tab.7 Lines 513-520 in the revision**), adding image-wise attention **improves per-frame feature learning** compared with using only multi-modal attention, particularly for sparse images inputs, and significantly **strengthens 2D–3D fusion**. In addition, point-wise attention which fuses local **geometric features based on Euclidean distances between points** in the point cloud, further boosting final performance.
>
> | Methods                              | In-the-wild Fashion 4 Views | In-the-wild Fashion 16 Views | Public & Smartphone 4 Views | Public & Smartphone 16 Views |
> | ------------------------------------ | --------------------------- | ---------------------------- | --------------------------- | ---------------------------- |
> | LHM++ w/o Image/Point-wise Attention | 21.669                      | 21.977                       | 27.403                      | 27.655                       |
> | LHM++ w/o Image-wise Attention       | 21.735                      | 22.136                       | 27.656                      | 27.927                       |
> | LHM++ w/o Point-wise Attention       | 21.838                      | 22.230                       | 27.764                      | 28.129                       |
> | LHM++                                | **21.957**                  | **22.354**                   | **27.940**                  | **28.394**                   |
>
> Finally, the proposed lightwegiht 3D-aware human animation neural renderer further improve the visual quality on under-cosntrain area. As the figure shown in ( **[Appendix D.6, Fig. 11, Lines 1174-1187]** in the revision**)** , as our LHM++ directly learns a positional residual from the SMPL-X template, challenges remain in rendering loose-fitting clothing , and unseen part of image, The proposed neural renderer not only effectively addresses this issue, but also significantly enhance the overall rendering results.
>
> ---
>
> #### ***Contribution for Efficiency Improvement***
>
> Another technical contribution is the improved efficiency compared with LHM’s architecture; we provide a more detailed analysis in **Weakness 3**.

---

> ### Author Response · Authors · 2025-11-21
> **Official Comment by Authors-C**
>
> ***W3: More detailed Complexity analysis & Efficiency technique contribution.***
>
> We'd love to give you a more detailed complexity analysis. First of all, our model is specifically designed for sparse image inputs, where the number of input images typically ranges from 1 to 16. Although we do not target at reducing the **theoritic** **computing complexity** in self-attention, we efficiently reduce the number of input tokens for self-attention, yielding a significant efficiency improvement over LHM while **preserving** the model's **performance**.
>
>
>
> Figure 5(d) in the paper provides a **qualitative** analysis of model efficiency. The reduction in compute tokens is mainly due to 3D-token grid pooling and 2D (image) token merging in the global attention stage. For **1–4 view** inputs, **3D tokens dominate**, so compressing them has the greatest impact on inference efficiency. As the number of **views** **increases (4–16)**, the share of image tokens grows, **making image-token compression increasingly effective** at speeding up inference.
>
>
>
> The following table ( **[Tab. 4 Lines 475-482]** in the revision**)** provides a **quantitative** analysis of model efficiency and clearly demonstrates the advantages of our pipeline over LHM. LHM requires expensive preprocessing (e.g., foreground parsing and separate extraction of body and head features), so for a fair comparison we report only the inference time for the multi-modality attention stage. Because LHM feeds all 2D and 3D tokens directly into the attention module, its attention computation is **extremely costly** and does **not scale well with the number of 3D tokens**. By contrast, our encoder–decoder design reduces redundant 2D and 3D tokens, enabling efficient scaling of the 3D token count. Under the same number of 3D points, our PIT is approximately **70–100× faster** than LHM.
>
> | # Points | 1 view LHM-0.7B | 1 view LHM++ (M) | 4 views LHM-0.7B | 4 views LHM++ (M) | 8 views LHM-0.7B | 8 views LHM++ (M) | 16 views LHM-0.7B | 16 views LHM++ (M) |
> | -------- | --------------- | ---------------- | ---------------- | ----------------- | ---------------- | ----------------- | ----------------- | ------------------ |
> | 40 K     | 5.16 s          | **0.55 s**       | 9.41 s           | **0.74 s**        | 15.23 s          | **1.01 s**        | 32.32 s           | **1.74 s**         |
> | 80 K     | 19.47 s         | **0.69 s**       | 25.73 s          | **0.88 s**        | 35.36 s          | **1.19 s**        | 59.25 s           | **1.95 s**         |
> | 160 K    | 74.31 s         | **0.79 s**       | 85.58 s          | **1.00 s**        | 102.44 s         | **1.31 s**        | 140.67 s          | **2.13 s**         |
>
>
>
> Furthermore, we include a new table (**[Tab. 5 Lines 486-494]** in the revision**)** showing the number of 3D tokens at each encoder layer of the proposed PIT, which clearly illustrates the dynamic changes in 3D token counts.
>
> | Operator                | 40K   | 80K   | 160K  |
> | ----------------------- | ----- | ----- | ----- |
> | Grid Pooling 1          | 16778 | 22380 | 24377 |
> | Point-Image Attention 1 | 16778 | 22380 | 24377 |
> | Grid Pooling 2          | 5547  | 6036  | 6441  |
> | Point-Image Attention 2 | 5547  | 6036  | 6441  |
> | Grid Pooling 3          | 1493  | 1625  | 1686  |
> | Point-Image Attention 3 | 1493  | 1625  | 1686  |

---

> ### Author Response · Authors · 2025-11-21
> **Official Comment by Authors-D**
>
> ***W4: The visualization of reconstruction avatar in canonical space.***
>
>
>
> Thanks for your suggestion. We added a figure illustrating human reconstructions in the canonical space (12 cases). ([**Appendix D.7, Lines 1219**], [**Appendix D.7, Fig.14 Lines 1242-1295, the 24 page**])
>
> ---
>
>
>
> ***W5: Missing Some References.***
>
> We sincerely appreciate your valuable suggestion. We have added citations to these works in （**[Tab. 1, Lines 117-132]，[Lines 147-149]**）
>
> ---
>
>
>
> ***Q1: What is main difference between LHM and LHM++ except for architecture details.***
>
> We adapted LHM to accept sparse images — a **nontrivial** task because the  original architecture’s **computational cost** becomes **prohibitive** as the  number of inputs increases. LHM++ introduces technical improvements that reduce **redundancy** and enable **efficient information fusion**, making it  well suited for sparse-image inputs. For implementation details, see our responses to ***W2-W3***.
>
> ---
>
>
>
> ***Q2: Why the gain of dense view input is marginal ?***
>
> We have responded to the question in ***W1-2***
>
> ---
>
>
>
> ***Q3: Failure Casaes.***
>
> Thank your for the suggestion. We have added the examples of failure cases in [**Appendix D.5, Lines 1065-1068**], [**Appendix D.5, Fig.9 Lines 1080-1095**].
>
> ---
>
>
>
>
>
> We welcome any further comments, questions, and discussions from you.

---

> ### Author Response · Authors · 2025-11-26
> **Official Comments by Authors**
>
> Dear Reviewer,
>
> We sincerely appreciate your valuable suggestions. Since only a few days remain in the discussion period, and to ensure we have time to provide additional materials if needed, could you please let us know whether our responses have addressed your concerns?
>
> We welcome any further questions and will respond promptly :).
>
> Best regards,
>
> LHM++ Authors

---

### Official Review · Reviewer_ckeM · 2025-10-27

**Soundness:** 3
**Presentation:** 3
**Contribution:** 2
**Rating:** 6
**Confidence:** 4

**Summary:**

The paper proposes LHM++, a network that reconstructs an animatable digital avatar in a feed-forward pass taking arbitrary number of images as inputs. It reduces the time cost of LHM by adopting token "merge" and "unmerge" operations. The token "merge" operation merges similar image tokens. By reducing the number of tokens, it speeds up the attention computations.

Another contribution is the neural renderer. Instead of rendering the predicted Gaussian directly, it renders the feature map in 2D and uses a DPT head to predict from the 2D feature map.

The modification in the number of tokens significantly speeds up the inference with more image inputs and reduces the memory cost. Meanwhile, the method outperforms the existing methods in terms of loose clothes animation due to the neural renderer.

**Strengths:**

* The paper demonstrates the ability to animate loose clothes and generalization, which is challenging in human rendering.
* With "merge" and "unmerge", the model runs much faster than LHM with a lower cost in memory.
* The paper is clearly written and highlights the contributions.

**Weaknesses:**

* The paper claims that in LHM, the time complexity of self-attention operations scales quadratically with the number of image tokens (and thus with the number of input images). Meanwhile, as the number of input images increases, image tokens begin to dominate the attention computation. Although the proposed “merge” and “unmerge” operations help reduce memory and computational overhead, the overall self-attention complexity remains quadratic with respect to the number of images. These operations only reduce the time cost by a constant factor.

* The proposed LHM++ is presented as an improvement over LHM; however, it is unclear where this improvement originates. The main contributions of the paper appear to be (1) the PIT block with the “token merge” mechanism and (2) the neural renderer. Since the “merge” and “unmerge” operations inevitably introduce information loss, they are more likely to degrade rather than enhance visual quality. Additionally, the paper’s ablation study in the appendix shows that the neural renderer only marginally improves PSNR and SSIM. I would appreciate it if the authors could clarify in more detail where the performance gain from LHM to LHM++ in Table 3 comes from.

**Questions:**

Please see weaknesses.

---

> ### Author Response · Authors · 2025-11-21
> **Official Comment by Authors-A**
>
> ***To Reviewer #ckeM***
>
>
>
> We thank the reviewer for the appreciation of the **generalization of model**, **lower cost** in memory and clearly written of LHM++. We address the weaknesses and questions point-by-point below.
>
>
>
> ---
>
>
>
> ***W1: Clarifying the Complexity Reduction***
>
>
>
> We appreciate the reviewer’s observation regarding the theoretical quadratic complexity of self-attention. Indeed, our work does **not aim to alter the asymptotic complexity bound** of the self-attention mechanism itself, but we develop a more flexible method that significantly reduces practical computational cost while preserving performance with sparse image inputs.
>
> We would like to examine whether it is **fair** to assess our model’s efficiency by comparing the proposed architecture to the prior work LHM **under some settings**, reporting both the runtime improvements and the performance achieved by LHM++.
>
> We conducted a comprehensive model-efficiency analysis (**[Lines 467-485], [Tab. 4 Lines475-482] in the revision).** The following table provides a **quantitative comparison** that highlights the advantages of our pipeline over LHM.
>
> | # Points | 1 view LHM-0.7B | 1 view LHM++ (M) | 4 views LHM-0.7B | 4 views LHM++ (M) | 8 views LHM-0.7B | 8 views LHM++ (M) | 16 views LHM-0.7B | 16 views LHM++ (M) |
> | -------- | --------------- | ---------------- | ---------------- | ----------------- | ---------------- | ----------------- | ----------------- | ------------------ |
> | 40 K     | 5.16 s          | **0.55 s**       | 9.41 s           | **0.74 s**        | 15.23 s          | **1.01 s**        | 32.32 s           | **1.74 s**         |
> | 80 K     | 19.47 s         | **0.69 s**       | 25.73 s          | **0.88 s**        | 35.36 s          | **1.19 s**        | 59.25 s           | **1.95 s**         |
> | 160 K    | 74.31 s         | **0.79 s**       | 85.58 s          | **1.00 s**        | 102.44 s         | **1.31 s**        | 140.67 s          | **2.13 s**         |
>
> LHM requires expensive preprocessing (e.g., foreground parsing and separate extraction of body and head features). To make the comparison fair, we report only **the inference time for the multi-modal attention stage**. Because LHM feeds all 2D and 3D tokens directly into the attention module, its attention computation is **very costly** and does not scale well with the number of 3D tokens. By contrast, our encoder–decoder design reduces redundant 2D and 3D tokens, enabling efficient scaling as the 3D token count increases.
>
> Under the same number of 3D points, our PIT is approximately **70–100×** faster than LHM, clearly demonstrating the efficiency gains of the proposed method.
>
> The remaining question is whether this speedup preserves performance. **Table 3 in the paper** presents comparative experiments showing that our model not only maintains **efficient** inference but also achieves **better** results. However, as noted in your second concern that it is unclear how the model achieves these improvements. We now address **Weakness 2**.
>
>
>
> ---

---

> ### Author Response · Authors · 2025-11-21
> **Official Comment by Authors-B**
>
> ***W2: the Incremental Gain in LHM++***
>
> Now, we provide detailed analysis what the performance gain from LHM++.
>
> Because an 3D avatar is represented by a set of sampled points, capturing finer model details requires a sufficient number of points. However, simply increasing the number of query points does not improve model performance. A key challenge remains: **how can we more effectively fuse a large set of 3D points with 2D image information**? To investigate this, we ran comparative experiments with LHM while varying the number of 3D tokens using  **8-view input**. the results are reported in the following table (**[Tab.6 Lines 501-507]** in the revision) .
>
> | Methods   | 40 K PSNR  | 40 K Time  | 80 K PSNR  | 80 K Time  | 160 K PSNR | 160 K Time |
> | --------- | ---------- | ---------- | ---------- | ---------- | ---------- | ---------- |
> | LHM-0.7B  | **21.761** | 24.76 s    | 21.803     | 49.78 s    | 21.796     | 110.37 s   |
> | LHM++ (M) | 21.735     | **1.86 s** | **22.124** | **1.98 s** | **22.208** | **2.09 s** |
>
> The results show that LHM yields only marginal gains as the number of 3D tokens increases. We attribute this to the need to learn **extremely large attention maps**. For example, attention matrices can contain **200K * 200K** (160K for 3D tokens, 40K for 2D tokens in LHM setting) configurations with 160K 3D tokens and 8 views image tokens. Such redundant tokens **hinder** learning and make **fusion inefficient**. In contrast, the Encoder-Decoder PIT architecture **reduces redundant 3D and 2D tokens** and **substantially improves fusion efficiency and overall performance**.
>
>
>
> To further analyze the improvements introduced by PIT, we conducted an ablation study on each attention module. As shown in the following table (**Tab.7 Lines 513-520 in the revision**), adding image-wise attention **improves per-frame feature learning** compared with using only multi-modal attention, particularly for sparse images inputs, and significantly **strengthens 2D–3D fusion**. In addition, point-wise attention which fuses local **geometric features based on Euclidean distances between points** in the point cloud, further boosting final performance.
>
> | Methods                              | In-the-wild Fashion 4 Views | In-the-wild Fashion 16 Views | Public & Smartphone 4 Views | Public & Smartphone 16 Views |
> | ------------------------------------ | --------------------------- | ---------------------------- | --------------------------- | ---------------------------- |
> | LHM++ w/o Image/Point-wise Attention | 21.669                      | 21.977                       | 27.403                      | 27.655                       |
> | LHM++ w/o Image-wise Attention       | 21.735                      | 22.136                       | 27.656                      | 27.927                       |
> | LHM++ w/o Point-wise Attention       | 21.838                      | 22.230                       | 27.764                      | 28.129                       |
> | LHM++                                | **21.957**                  | **22.354**                   | **27.940**                  | **28.394**                   |
>
> Benefiting from the carefully designed modules in PIT, LHM++ achieves both high efficiency and strong performance.
>
>
>
> ---
>
> We welcome any further comments, questions, and discussions from you.

---

> > ### Comment · Reviewer_ckeM · 2025-11-25
> >
> > Thanks for the clarifications!
> >
> > I have one remaining question regarding the performance gain over LHM++. I now understand that the three types of attentions, i.e. image/point-wise, image-wise and point-wise attention, combined contribute to the performance. However, I am still unsure about the role of the merging/unmerging operations in fusion efficiency. Specifically, do these operations enhance fusion efficiency in a way that improves rendering quality, rather than degrading the rendering quality?

---

> > > ### Author Response · Authors · 2025-11-26
> > > **Official Comment by Authors on Token Merging**
> > >
> > > Thank you for your reply. We will further analyze the effectiveness of the **merge** and **unmerge** operators.
> > >
> > >
> > >
> > > To further analyze the **merge** and **unmerge** operator in global image-point attention, we conducted both qualitative and quantitative experiments that demonstrate how performance varies with different merge ratios **r** and numbers of views.
> > >
> > > ----
> > >
> > >
> > >
> > > The figure in  [**Appendix F.1, Fig.17 Lines 1425-1454** ]  illustrates qualitative results showing how visual quality changes with different token compression ratios across sparse-view inputs. The figure indicates that for **1~4 views**,  the **high merge ratio r > 0.5** significantly degrades visual quality, whereas for 8~16 views, larger merge ratios **r ∈ [0.5, 0.75]** do not obviously reduce performance. We attribute this to **the increased redundancy of tokens** **as the number of views grows**, and the our model de facto does not require all tokens.
> > >
> > > ---
> > >
> > > The following table [**Appendix F.1, Tab.15 Lines 1481-1489** ] presents a quantitative ablation of image-token merging. The table reports PSNR averaged on both public and smartphone video sequences. It shows that applying token merging with moderate ratios r ∈ [0.5, 0.75] for sparse-view inputs 4~16 yields metrics that are **similar to, or slightly better than**  the **no-merge baseline r = 0**.  Moreover, as we use a neural renderer to enhance the final renderings, high merge ratios do not significantly degrade quantitative metrics but do cause a noticeable drop in visual quality, See in  [**Appendix F.1, Fig.17 Lines 1425-1454** ]
> > >
> > > | Input Views |   r=0.0    |   r=0.25   |   r=0.5    | r=0.75 | r=0.90 | r=0.95 |
> > > | :---------: | :--------: | :--------: | :--------: | :----: | :----: | :----: |
> > > |  view = 1   | **27.762** |   27.735   |   27.733   | 27.644 | 27.525 | 27.238 |
> > > |  view = 4   |   27.942   | **27.942** |   27.940   | 27.937 | 27.893 | 27.662 |
> > > |  view = 8   |   28.140   | **28.153** |   28.147   | 28.135 | 27.903 | 27.735 |
> > > |   view=16   |   28.391   | **28.394** | **28.394** | 28.391 | 28.358 | 28.206 |
> > >
> > > ---
> > >
> > > Notably,  **the similar findings were reported**  the previous work, [Token Merging for Stable Diffusion]()  (https://arxiv.org/pdf/2303.17604) [Please see in Table.4 Page 4]. **In that study, token merging produced slightly better photometric results than the baseline methods.**
> > >
> > > -----
> > >
> > > **Reference**:
> > >
> > > [1] Token Merging for Stable Diffusion, CVPR Workshop 2023, Oral.
> > >
> > > ---
> > >
> > > I hope our comments can address your concerns.
> > >
> > > Again, we welcome any further comments or questions :).

---

> > > > ### Comment · Reviewer_ckeM · 2025-11-26
> > > >
> > > > Thanks for the clarifications. I encourage the authors to include the analysis on the new proposed components in the revised paper. Given the comprehensive analysis on the proposed components and the improvement over LHM, I would like to raise the score to 8.

---

> > > > > ### Author Response · Authors · 2025-11-27
> > > > > **Official Comment by Authors**
> > > > >
> > > > > We sincerely appreciate your positive vote. Your suggestions have greatly improved the comprehensiveness of our paper. We have added analyses of the newly proposed components in the revised manuscript. We retained the original figure–text ordering from the first revision and have temporarily appended the new material to the last page. Once all rebuttals are finalized, we will integrate the added content into the most appropriate locations in the manuscript and provide a detailed change log in the updated comments.
> > > > >
> > > > > Thank you again for your valuable feedback.

---

### Official Review · Reviewer_MuBq · 2025-10-29

**Soundness:** 3
**Presentation:** 3
**Contribution:** 3
**Rating:** 6
**Confidence:** 3

**Summary:**

This paper presents LHM++: AN EFFICIENT LARGE HUMAN RECONSTRUCTION MODEL FOR POSE-FREE IMAGES TO 3D. Overall, the proposed method is well-motivated, and the experimental results seems good.

**Strengths:**

•	The paper is well-written with a logical structure that makes the technical contributions easy to follow.
•	The proposed  framework is reasonable and well-justified. The experimental results convincingly demonstrate the effectiveness of the approach across different scenarios.
•	 The demo videos are excellent supplementary materials.

**Weaknesses:**

Could you please give a discussion about the diffirence with 3D generation model, such like CLAY (Rodin). I wonder can we use the Rodin to perform 3D avatar generation and then perform auto-rigging such as Mixamo?

**Questions:**

please see the weaknesses

---

> ### Author Response · Authors · 2025-11-21
> **Official Comment by Authors**
>
> #### ***To Reviewer #Bubq:***
>
> We thank the reviewer for your appreciation of the **reasonable, well-justified** framework, **effectiveness** and **excellent Videos** result of LHM++. We address your the weakness below.
>
> ---
>
> #### ***Weakness: Discussion and Comparison with general 3D generation model.***
>
> we conducted comparative experiments against general image-to-3D methods. We chose Hunyuan3D-2.5 as the baseline because of its state-of-the-art performance.
>
> Following your suggestion, after reconstructing a 3D model from the input-view avatar we use Mixamo to auto‑rig the models, turning any input image into an animatable avatar. As shown in ([**Appendix D.4, Fig. 8, Lines 1039-1060**]), rigging a 3D digital human imposes strict requirements on the input pose. While Hunyuan3D-2.5 can recover the overall structure, it often fails to produce a riggable geometry: in the "Reset" panel, points on the arms and torso are difficult to distinguish, making the avatar hard to animate. Its generated textures also frequently look unnatural or contain artifacts. By contrast, LHM++ produces natural, high-fidelity avatars in canonical space.
>
> We add the discussion on ([**Appendix D.4 Lines 1028-1038**], [**Appendix D.4, Fig.8 Lines 1039-1060**]) in the revison.
>
> ---
>
> We welcome any further comments, questions, and discussions from you.

---

> ### Author Response · Authors · 2025-11-26
> **Official Comments by Authors**
>
> Dear Reviewer,
>
> We sincerely appreciate your valuable suggestions. Since only a few days remain in the discussion period, and to ensure we have time to provide additional materials if needed, could you please let us know whether our responses have addressed your concerns?
>
> We welcome any further questions and will respond promptly :).
>
> Best regards,
>
> LHM++ Authors

---

### Official Review · Reviewer_3ReR · 2025-11-01

**Soundness:** 3
**Presentation:** 3
**Contribution:** 2
**Rating:** 4
**Confidence:** 4

**Summary:**

This paper proposes LHM++, a feed-forward model to generate 3D human avatars from casually captured images. At the core of the method is a Encoder-Decoder Point-Image Transformer (PIT) module to fuse 3D and 2D features, which are decoded into 3D Gaussian parameters. The authors conducted experiments on different dataset to verify the effectiveness of the proposed method.

**Strengths:**

The paper is easy to follow.

Synthesizing 3D/4D humans from images is an interesting task with practical applications.

The method is technically sound by leveraging a multimodal transformer architecture to fuse 3D and 2D feature for 3D Gaussian generation.

**Weaknesses:**

Limited technical contribution. This paper is an extension for LHM, and the main difference is that LHM++ replaces the MBHT with PIT mode. However, both MBHT and PIT fuse 3D and 2D features for 3D Gaussian prediction. Does the LHM support multiple image processing by fusing multiple images using MBHT architecture? Why is the PIT required, and how does it outperform MBHT?

The paper proposes that the PIT architecture improves the results, whereas the results in Tab. 10 suggest that the number of 3D geometric points has a bigger impact on the results, i.e., for 40K points, LHM-0.7B even performs better. It’s not clear whether the PIT architecture or the number of query points improves the results.


The paper proposes DPT-head as the final renderer. In this case, why is the 3DGS representation required? Is it possible to just predict the SMPL offsets for LBS instead of the full 3DGS parameters?

**Questions:**

How does the method decouple the belongings (e.g., the bag in Fig. 2) and human clothing?

Details about the implementation. Are the Gaussian rendering and neural rendering jointly trained? The Eq. 8 loss is not clear. Are both the RGB and perception loss applied for Gaussian rendering and neural rendering?

---

> ### Author Response · Authors · 2025-11-21
> **Official Comment by Authors-A**
>
> ***To Reviewer #3ReR:***
>
> We thank the reviewer for your appreciation of the **the paper is easy to follow, an interesiting task with practical application**, and **technically Sound** of LHM++. We address your the weaknesses and questions below.
>
>
> ---
>
> ***W1: Does LHM support sparse images input and why is the PIT required?***
>
> Yes, LHM can handle sparse image inputs; however, its efficiency degrades significantly in sparse-view scenarios.
>
>
>
> The following table ( **[Tab. 4 Lines475-482]** in the revision**)** provides a **quantitative** analysis of model efficiency and clearly demonstrates the advantages of PIT over LHM's *MBHT*.
>
>
>
> | # Points | 1 view LHM-0.7B | 1 view LHM++ (M) | 4 views LHM-0.7B | 4 views LHM++ (M) | 8 views LHM-0.7B | 8 views LHM++ (M) | 16 views LHM-0.7B | 16 views LHM++ (M) |
> | -------- | --------------- | ---------------- | ---------------- | ----------------- | ---------------- | ----------------- | ----------------- | ------------------ |
> | 40 K     | 5.16 s          | **0.55 s**       | 9.41 s           | **0.74 s**        | 15.23 s          | **1.01 s**        | 32.32 s           | **1.74 s**         |
> | 80 K     | 19.47 s         | **0.69 s**       | 25.73 s          | **0.88 s**        | 35.36 s          | **1.19 s**        | 59.25 s           | **1.95 s**         |
> | 160 K    | 74.31 s         | **0.79 s**       | 85.58 s          | **1.00 s**        | 102.44 s         | **1.31 s**        | 140.67 s          | **2.13 s**         |
>
> Since LHM requires expensive preprocessing (e.g., foreground parsing and separate extraction of body and head features), for a fair comparison we report only the **inference time for the multi-modality attention stage**. Because LHM feeds all 2D and 3D tokens directly into the attention module, its attention computation is **extremely costly** and does **not scale well with the number of 3D tokens**.
>
>
>
> By contrast, our encoder–decoder design significantly reduces redundant 2D and 3D tokens, enabling efficient scaling of the 3D token count. Under the same number of 3D points, our PIT is approximately **70–100×** faster than LHM.
>
>
>
> The experiment above focuses on the **efficiency** **aspect** for adopting PIT. Beyond **substantially outperforming** LHM’s MBHT in efficiency, PIT also delivers improved overall performance compared to the original LHM. This naturally raises the question of **where those performance gains come from;** we address that in ***Weakness 2 (W2)***.

---

> ### Author Response · Authors · 2025-11-21
> **Official Comment by Authors-B**
>
> ***W2: Performacne Gain issues about the number of points & Clarifying the table.10 & How dose PIT outperform MBHT.***
>
> We appreciate the reviewer's comment on Table. 10 (**Now in Tab. 12**) and it helped us strengthen the paper.
>
>
>
> Indeed, the number of 3D points is critically important for model performance: a larger point count enables the model to capture **finer geometric** details. However, this **does not imply that increasing the number of points invariably leads to better learning outcomes**. Under conditions with more point input, the **key** to further performance gains lies in **effectively fusing sparse image inputs information** with the 3D point representations.
>
> To analyze this, we ran comparative experiments with LHM while varying the number of 3D tokens using an 8-view input. the results are reported in the following table (**[Tab.6 Lines 501-507]** in the revision**) .**
>
>
>
> | Methods   | 40 K PSNR  | 40 K Time  | 80 K PSNR  | 80 K Time  | 160 K PSNR | 160 K Time |
> | --------- | ---------- | ---------- | ---------- | ---------- | ---------- | ---------- |
> | LHM-0.7B  | **21.761** | 24.76 s    | 21.803     | 49.78 s    | 21.796     | 110.37 s   |
> | LHM++ (M) | 21.735     | **1.86 s** | **22.124** | **1.98 s** | **22.208** | **2.09 s** |
>
> The results show that LHM yields only marginal gains as the number of 3D tokens increases. We attribute this to the need to learn **extremely large attention maps**. For example, attention matrices can contain **200K * 200K** (160K for 3D tokens, 40K for 2D tokens in LHM setting) configurations with 160K 3D tokens and 8 views image tokens. Such redundant tokens **hinder** learning and make **fusion inefficient**. In contrast, the Encoder-Decoder PIT architecture **reduces redundant 3D and 2D tokens** and **substantially improves fusion efficiency and overall performance**.
>
>
>
> To further analyze the improvements introduced by PIT, we conducted an ablation study on each attention module. As shown in the following table (**Tab.7 Lines 513-520 in the revision**), adding image-wise attention **improves per-frame feature learning** compared with using only multi-modal attention, particularly for sparse images inputs, and significantly **strengthens 2D–3D fusion**. In addition, point-wise attention which fuses local **geometric features based on Euclidean distances between points** in the point cloud, further boosting final performance.
>
> | Methods                              | In-the-wild Fashion 4 Views | In-the-wild Fashion 16 Views | Public & Smartphone 4 Views | Public & Smartphone 16 Views |
> | ------------------------------------ | --------------------------- | ---------------------------- | --------------------------- | ---------------------------- |
> | LHM++ w/o Image/Point-wise Attention | 21.669                      | 21.977                       | 27.403                      | 27.655                       |
> | LHM++ w/o Image-wise Attention       | 21.735                      | 22.136                       | 27.656                      | 27.927                       |
> | LHM++ w/o Point-wise Attention       | 21.838                      | 22.230                       | 27.764                      | 28.129                       |
> | LHM++                                | **21.957**                  | **22.354**                   | **27.940**                  | **28.394**                   |
>
> Benefiting from the carefully designed modules in PIT, LHM++ achieves both **high efficiency** and **strong performance**.
>
> ---
>
> In summary, with respect to the mentionedWeaknesses 1 & 2, the above experiments show that the encoder–decoder PIT is **not merely an extension** of LHM but an **efficient**, **carefully designed** architecture for sparse image inputs that delivers a **significant** speedup as well as improved results. This is why we require PIT.

---

> ### Author Response · Authors · 2025-11-21
> **Official Comment by Authors-C**
>
> ***W3: The Necessity of 3DGS.***
>
> Following the suggestion, we conducted an ablation in which we deformed only the points sampled from SMPL-X (modifying their positions) and rendered features via geometric rasterization implemented with PyTorch3D, followed by rendering with the DPT head — i.e., omitting the 3DGS representation used in our full pipeline.
>
>
>
> Figure 13 in [**Appendix D.6, Fig.13 Lines 1206-1216**] shows the comparison using the suggested alternative. Relying solely on point offsets, without an explicit 3DGS representation, causes the DPT head to **struggle to reproduce** input-image context, resulting in **high-frequency artifacts, blurring**, and **convergence difficulties**. By contrast, including an explicit 3DGS representation lets the DPT head converge faster and substantially improves overall visual quality.
>
>
>
> ---
>
> ***Q1: How to decouple the belongings of input image during inference.***
>
> Sorry for the confusion. Our RGB and mask supervision uses a foreground mask that excludes belongings (e.g., bags). Consequently, the feedforward model automatically learns to ignore or remove these belongings from the input image by self-attention mechanism. An example of this ground-truth RGB supervision appears in [**Appendix D.6, Fig.13 Lines 1206-1216**, **GT panels**] .
>
> ---
>
>
>
> ***Q2: Training loss in eq.8.***
>
> Yes, the photometric losses apply to both the Gaussian-splat-rendered and the neural-rendered images. Actually, we described the RGB and mask loss terms (**Lines 345-347**] ) but did not clearly explain the LPIPS loss; we have added a sentence in (**Lines 347-348**] ) the revision to clarify the LPIPS term.
>
>
>
> ---
>
> We welcome any further comments, questions, and discussions from you.

---

> ### Author Response · Authors · 2025-11-26
> **Official Comments by Authors**
>
> Dear Reviewer,
>
> We sincerely appreciate your valuable suggestions. Since only a few days remain in the discussion period, and to ensure we have time to provide additional materials if needed, could you please let us know whether our responses have addressed your concerns?
>
> We welcome any further questions and will respond promptly :).
>
> Best regards,
>
> LHM++ Authors

---

### Author Response · Authors · 2025-11-21
**Official Comment by Authors**

Dear Reviewers,



We sincerely thank you for your valuable suggestions, which have helped us improve and strengthen the paper. In this revision, we have updated the following items:

- We have updated Table 1 to provide a more comprehensive survey of state-of-the-art 3D human reconstruction methods. （**[Tab. 1, Lines 117-132]，[Lines 147-149]**）

- We provide more detailed descriptions of the training loss to avoid potential confusion. （ **Lines 347-348**）
- We corrected a testing-time typo for LHM: we had copied their reported value, but they measured inference time on an RTX 4090 while our tests were run on a single NVIDIA A100. Accordingly, we updated the value from "3.17 s" to "5.73 s" (**Tab. 3, Line 423**).
- We have added a more detailed efficiency analysis in the main text to enable reviewers to clearly observe the performance gap between our proposed framework and LHM in terms of computational efficiency. This analysis includes two new tables: Table 4 and Table 5. **Table 4** reports the inference time required by LHM and our method to fuse 2D and 3D tokens (excluding image tokenization and preprocessing) under varying input configurations and different numbers of geometric sampling points. **Table 5** illustrates the reduction in the number of 3D tokens across the encoder's layers of our PIT framework. (**[Lines 467-485], [Tab. 4 Lines475-482]**, **[Tab. 5 Lines486-494]** )
- We added a detailed study on the effectiveness of multi-modal fusion, including Table 6, which shows our model can efficiently fuse large numbers of 3D tokens and achieve overall quality improvements, whereas LHM struggles to fuse 2D and 3D tokens efficiently when the 3D token count is high. (**[Lines 500-512], [Tab.6 Lines 501-507]**)
- We also presented a comprehensive ablation study of the attention modules in our encoder–decoder PIT, analyzing the contributions of image-wise and point-wise attention to the performance gains. (**[Lines 513-528], [Tab.7 Lines 513-520]**)
- We discussed general image-to-3D methods (e.g., Hunyuan3D and Rodin) for avatar generation, as well as the use of Mixamo for automatic rigging, and include a comparative experiment evaluating our approach against these pipelines. ([**Appendix D.4 Lines 1028-1038**], [**Appendix D.4, Fig.8 Lines 1039-1060**])
- We added a figure illustrating failure cases of our method. ([**Appendix D.5, Lines 1065-1068**], [**Appendix D.5, Fig.9 Lines 1080-1095**])
- We corrected errors in the Model Efficiency table: the LHM efficiency value was computed using 40K points, not 160K. We replaced "160 K" with "40K" in the "# Points" column and added a new row reporting LHM's efficiency with 160K points. ([**Appendix D.6, Tab. 12, Lines 1102-1103**])
- We conducted an ablation study to verify that the 3DGS representation plays a important role in 3D-aware Neural Rendering. ([**Appendix D.6, Lines 1173-1204**], [**Appendix D.6, Fig.13 Lines 1206-1216**])
- We added a figure illustrating human reconstructions in the canonical space (12 cases). ([**Appendix D.7, Lines 1219-1221**], [**Appendix D.7, Fig.14 Lines 1242-1295, the 24 page**])

---

***News Update***

 We have updated the following items:
-  [Nov. 26th] we add both quantitative and qualitative results for the ablation study of image-token merging   ([**Appendix F, Lines 1412-1424**], [**Appendix F.1, Fig.17 Lines 1426-1451**], [**Appendix F.1, Tab. 15 Lines 1481-1489**], )

---

For the specific concerns and questions raised by the reviewers, we have provided detailed and structured point-by-point responses under the corresponding comments.


Sincerely,

LHM++ Authors

---

> ### Author Response · Authors · 2025-11-25
> **Official Comment by Authors**
>
> Dear Reviewers,
>
> Following your suggestions, we updated the main paper and Appendix on Nov 21. The newly added results in the revision are highlighted in red.
>
> These updates include additional analyses, figures, and tables, and do not affect our original conclusions.
>
> Best,
>
> Authors

---

### Author Response · Authors · 2025-11-29
**Rebuttal Summary and Request for Consideration**

Dear Area Chair,

We sincerely thank the anonymous reviewers for their time, thoughtful  feedback, and voluntary service, and we greatly appreciate your  coordination in overseeing our submission.

---

This paper was reviewed by four expert reviewers. During the rebuttal  phase, Reviewer ckeM raised two questions: one requesting clarification  on the time complexity of LHM++, and another concerning its incremental  gains. After we submitted our responses, the reviewer asked an  additional question about the effects of image-token merging; we  provided supporting figures and tables in a follow-up reply. **As a result**, the  reviewer **increased their score from 6 to 8**. This **update was completed at approximately 21:29 UTC on November 26, 2025**, which **predates** the public disclosure of the information leak (widely spread  **between 14:00 and  16:00 UTC on November 27, 2025**).

Unfortunately, because the leaked information had already spread, the  other three reviewers were unable to submit final scores after our  rebuttal, despite we submitted detailed responses for all of their weaknesses/questions in the  corresponding comment sections.

---

For your reference, we summarize our rebuttal efforts, including our  responses to the reviewers’ concerns, the additional clarifications and  revisions we provided, and the subsequent exchanges during the  discussion phase. In this revision, we have updated the following items:

- We have updated Table 1 to provide a more comprehensive survey of state-of-the-art 3D human reconstruction methods. （**[Tab. 1, Lines 117-132]，[Lines 147-149]**） **@ Reviewer nJbr**

- We provide more detailed descriptions of the training loss to avoid potential confusion. （ **Lines 347-348**）**@ Reviewer 3ReR**
- We corrected a testing-time typo for LHM: we had copied their reported value, but they measured inference time on an RTX 4090 while our tests were run on a single NVIDIA A100. Accordingly, we updated the value from "3.17 s" to "5.73 s" (**Tab. 3, Line 423**).
- We added a detailed efficiency analysis to the main text comparing our  framework and LHM. It includes Tab. 4, which reports inference time for fusing 2D and 3D tokens (excluding image tokenization and  preprocessing) across input configurations and geometric sampling  counts, and Tab. 5, which shows the reduction in 3D token counts across the encoder layers of our PIT framework. (**[Lines 467-485], [Tab. 4 Lines475-482]**, **[Tab. 5 Lines486-494]** ) **@ Reviewer (3ReR &  ckeM & njbr)**
- We added a detailed study on the effectiveness of multi-modal fusion, including Table 6, which shows our model can efficiently fuse large numbers of 3D tokens and achieve overall quality improvements, whereas LHM struggles to fuse 2D and 3D tokens efficiently when the 3D token count is high. (**[Lines 500-512], [Tab.6 Lines 501-507]**)  **@ Reviewer (3ReR &  ckeM & njbr)**
- We also presented a comprehensive ablation study of the attention modules in our encoder–decoder PIT, analyzing the contributions of image-wise and point-wise attention to the performance gains. (**[Lines 513-528], [Tab.7 Lines 513-520]**)   **@ Reviewer (3ReR &  ckeM & njbr)**
- We discussed general image-to-3D methods (e.g., Hunyuan3D and Rodin) for avatar generation, as well as the use of Mixamo for automatic rigging, and include a comparative experiment evaluating our approach against these pipelines. ([**Appendix D.4 Lines 1028-1038**], [**Appendix D.4, Fig.8 Lines 1039-1060**]) **@ Reviewer MuBq**
- We added a figure illustrating failure cases of our method. ([**Appendix D.5, Lines 1065-1068**], [**Appendix D.5, Fig.9 Lines 1080-1095**])   **@ Reviewer nJbr**
- We corrected errors in the Model Efficiency table: the LHM efficiency value was computed using 40K points, not 160K. We replaced "160 K" with "40K" in the "# Points" column and added a new row reporting LHM's efficiency with 160K points. ([**Appendix D.6, Tab. 12, Lines 1102-1103**])  **@ Reviewer 3ReR**
- We conducted an ablation study to verify that the 3DGS representation plays a important role in 3D-aware Neural Rendering. ([**Appendix D.6, Lines 1173-1204**], [**Appendix D.6, Fig.13 Lines 1206-1216**]) **@ Reviewer 3ReR**
- We added a figure illustrating human reconstructions in the canonical space (12 cases). ([**Appendix D.7, Lines 1219-1221**], [**Appendix D.7, Fig.14 Lines 1242-1295, the 24 page**])  **@ Reviewer njbr**

- [10.26]  we add both quantitative and qualitative results for the ablation study of image-token merging   ([**Appendix F, Lines 1412-1424**], [**Appendix F.1, Fig.17 Lines 1426-1451**], [**Appendix F.1, Tab. 15 Lines 1481-1489**] )   **@ Reviewer ckeM**

---

Thank you very much for your time and consideration. We would be  grateful if you could take into account the progress made during the  rebuttal, particularly the revised score from Reviewer ckeM and our  comprehensive responses to all reviewers, when evaluating our  submission.



Sincerely,

LHM++ Authors

---

### Meta-Review · Area_Chair_xjKS · 2026-01-07

**Summary:**

This submission proposes LHM++, a feed-forward system for reconstructing animatable 3D human avatars from one or multiple pose-free images. The key component is an encoder–decoder PIT module that fuses 2D image tokens with 3D point tokens and decodes to 3D Gaussian splats. Across the four reviews, the paper was generally regarded as well written, and reviewers acknowledged the practical value of fast inference. However, multiple reviewers viewed LHM++ as close to LHM in overall formulation, and the performance gain is also not significant. The authors have provided an evaluation on 40k, 80k, 160k points to indicate the gains as more points are sampled. However, the gain is also very subtle, raising the question of whether increasing the number of points is beneficial in practice. Taking all the evaluations and ablation tables into account, it is believed that the improvement mainly comes from the PIT module, which is basically a combo of existing techniques. Therefore, given the limited technical contribution and marginal gains, the final recommendation is Reject.

**Reviewer Concerns:**

Addressed: 1) Clarification of efficiency: The authors added detailed runtime tables and token-count analyses to support the claim that PIT reduces redundant tokens and improves practical efficiency relative to LHM. 2) ablations on the performance gains: The rebuttal includes additional ablations of attention components (image-wise / point-wise) to attribute performance improvements to PIT design. 3) Token merging ratio: The authors added results over merge ratios and view counts, clarifying their effects on final results. 4) Some minor issues: failure cases were added, and examples of the canonical reconstruction are provided.

Not fully addressed: 1) limited novelty as compared to LHM: More results are given, and the authors emphasized the inference efficiency, while it is still not clear what the novelty is regarding the core network design. 2) While the rebuttal provides extensive timing analyses and argues large practical speedups, reviewers noted that the attention complexity remains quadratic in token count.

**Reviewer Scores:**

Reviewer ckeM stated that the score should be raised from 6 to 8. The other three reviewers didn't provide further feedback explicitly. Reviewer 3ReR and nJbr might keep the initial rating since the major concern about the limited novelty is not fully addressed. Reviewer MuBq raised questions that are quite related to the core idea of this submission, so it is hard to predict the final response from this reviewer.

---

### Decision · Program_Chairs · 2026-01-26

Reject